# *In silico* Pathway Activation Network Decomposition Analysis (iPANDA) as a method for biomarker development

Ivan V. Ozerov[1,*], Ksenia V. Lezhnina[1,*], Evgeny Izumchenko[2], Artem V. Artemov[1], Sergey Medintsev[1], Quentin Vanhaelen[1], Alexander Aliper[1,3], Jan Vijg[4], Andreyan N. Osipov[1,3], Ivan Labat[5], Michael D. West[5], Anton Buzdin[1,3,6], Charles R. Cantor[7], Yuri Nikolsky[1,8], Nikolay Borisov[1,3,6], Irina Irincheeva[9], Edward Khokhlovich[10], David Sidransky[2], Miguel Luiz Camargo[10] & Alex Zhavoronkov[1,3,11]

Signalling pathway activation analysis is a powerful approach for extracting biologically relevant features from large-scale transcriptomic and proteomic data. However, modern pathway-based methods often fail to provide stable pathway signatures of a specific phenotype or reliable disease biomarkers. In the present study, we introduce the *in silico* Pathway Activation Network Decomposition Analysis (iPANDA) as a scalable robust method for biomarker identification using gene expression data. The iPANDA method combines precalculated gene coexpression data with gene importance factors based on the degree of differential gene expression and pathway topology decomposition for obtaining pathway activation scores. Using Microarray Analysis Quality Control (MAQC) data sets and pretreatment data on Taxol-based neoadjuvant breast cancer therapy from multiple sources, we demonstrate that iPANDA provides significant noise reduction in transcriptomic data and identifies highly robust sets of biologically relevant pathway signatures. We successfully apply iPANDA for stratifying breast cancer patients according to their sensitivity to neoadjuvant therapy.

[1] Pharmaceutical Artificial Intelligence Department, Insilico Medicine, Inc., Emerging Technology Centers, Johns Hopkins University at Eastern, B301, 1101 33rd Street, Baltimore, Maryland 21218, USA. [2] The Johns Hopkins University, School of Medicine, Department of Otolaryngology, Head and Neck Cancer Research, 1550 Orleans Street, Baltimore, Maryland 21231, USA. [3] Laboratory of Bioinformatics, D. Rogachev Federal Research and Clinical Center for Pediatric Hematology, Oncology and Immunology, Samory Mashela 1, Moscow 117997, Russia. [4] Department of Genetics, Albert Einstein College of Medicine, 1300 Morris Park Avenue, Bronx, New York 10461, USA. [5] BioTime, Inc., 1010 Atlantic Avenue, Alameda, California 94501, USA. [6] National Research Centre 'Kurchatov Institute', Centre for Convergence of Nano-, Bio-, Information and Cognitive Sciences and Technologies, 1, Akademika Kurchatova square, Moscow 123182, Russia. [7] Boston University, Department of Biomedical Engineering, 44 Cummington Street, Boston, Massachusetts 02215, USA. [8] Skolkovo Foundation, 5 Nobelya street, Skolkovo Innovation Centre, Mozhajskij region, Moscow 143026, Russia. [9] Nutrition and Metabolic Health group, Nestlé Institute of Health Sciences, CH-1015 Lausanne, Switzerland. [10] Novartis Institutes for BioMedical Research, 250 Massachusetts Avenue, Cambridge, Massachusetts 02139, USA. [11] The Biogerontology Research Foundation, 2354 Chynoweth House, Trevissome Park, Truro TR4 8UN, UK. * These authors contributed equally to this work. Correspondence and requests for materials should be addressed to I.V.O. (email: ivan@insilicomedicine.com) or to K.V.L. (email: ksenia@insilicomedicine.com) or to A.Z. (email: alex@insilicomedicine.com).

The application of novel supervised learning algorithms to large-scale transcriptomic data has the potential to transform conventional approaches for disease classification, personalized medicine and development of prognostic models. However, their use as a modality for clinical applications is hindered by several recognized challenges and limitations. One of the most relevant challenges in transcriptomic data analysis is the inherent complexity of gene network interactions, which remains a significant obstacle in building comprehensive predictive models. Moreover, high diversity of experimental platforms and inconsistency of the data coming from the various types of equipment—may also lead to the incorrect interpretation of the underlying biological processes. Although a number of data normalization approaches have been proposed over the recent years[1,2], it remains difficult to achieve robust results over a group of independent data sets even when they are obtained from the same profiling platform[3]. This may be explained by a range of biological factors, such as wide heterogeneity among individuals on the population basis, variance in the cell cycle stage of the cells used or a set of technical factors, such as sample preparation or batch variations in reagents.

Despite these challenges, various transcriptomic data analysis algorithms have been developed in both academic and commercial settings, and numerous attempts have been made to apply these algorithms clinically, particularly, for predicting patient response to various anti-cancer therapies[4–6]. Canonically, these approaches are intended to identify differentially expressed genes between groups of samples. Although this can lead to the identification of prospective genetic biomarkers and expression signature patterns of the process under study, it fails to capture subtle differences between samples that arise from dynamic interactions between genes at the level of signalling networks. Additionally, noise generated by variations in experimental protocols may further affect the ability of any approach to accurately detect the distinction between samples. To circumvent these limitations, a number of computational scoring platforms that can project gene expression data into a molecular signalling network have been proposed for integrative pathway analysis[7]. The major advantage of pathway-based methods is their capability to perform biologically relevant dimension reduction as a result of the analysis. However, despite significant advancements, current pathway-based methods are still imperfect in extrapolating the functional states of transcriptomes into the biological networks. Many popular pathway-based algorithms, such as Gene Set Enrichment Analysis (GSEA) and its extensions, rely solely on gene enrichment statistics, treating pathways as unstructured sets of genes[8]. Another group, including Signalling Pathway Impact Analysis (SPIA), Topology Gene-Set Analysis, and DEGraph, treats pathways as directed or undirected graphs representing networks of biochemical interactions at gene and protein levels[9–11]. Oncofinder algorithm represents a halfway approach, where information about pathway topology is used to assign activation or repression roles of particular genes in the pathway and then estimate its overall activation[12]. Although very helpful, these approaches cannot overcome other above-mentioned limitations, posing a need for development of the new large-scale analytical methodologies that infer complex transcriptomic changes more accurately into the network of biologically relevant signalling axes.

In this study, we suggest a novel method for large-scale transcriptomic data analysis called *in silico* Pathway Activation Network Decomposition Analysis (iPANDA). We demonstrate the performance of this method by using multiple paclitaxel breast cancer treatment data sets obtained from Gene Expression Omnibus (GEO)[13]. Breast cancer data was chosen for the analysis

as one of the most challenging in several ways. Since breast cancer has a high degree of intertumour and intratumoural heterogeneity, this cancer type is one of the most difficult in terms of outcome and treatment response prediction[14]. This is especially true for a group of tumours with poor prognosis and fewer number of effective treatments such as estrogen receptor negative breast cancer types (human epidermal growth factor receptor 2 (HER2)-positive and HER2-negative)[15]. Thus, traditional methods for transcriptomic data analysis may not be sufficient in this particular case. Breast cancer is also the second most common cancer in the US after skin cancer and second leading cause of cancer death in women after lung cancer[16]. Hence, there is an unmet need for development of new generation highly robust transcriptomic data analysis methods. Our study demonstrates that iPANDA is an effective tool for biologically relevant dimension reduction in transcriptomic data. Using neoadjuvant therapy pretreatment breast cancer data with known treatment outcome and receptor status (estrogen receptor and HER2), we show that iPANDA is capable of producing highly robust sets of pathway markers, which can be further used for stratification of samples into responder and non-responder groups.

## Results

**Overview of the iPANDA method.** Fold changes between the gene expression levels in the samples under investigation (tumour samples) and an average expression level of samples within the normal set is used as input data for the iPANDA algorithm. Since some genes may have a stronger effect on the pathway activation than others, the gene importance factor has been introduced. Several approaches of gene importance hierarchy calculation have been proposed during the last few decades[7]. The vast majority of these approaches aim to enrich pathway-based models with specific gene markers most relevant for a given study. While some of them use detailed kinetic models of several particular metabolic networks to derive importance factors[17], in others, gene importance is derived from the statistical analysis of the gene expression data obtained for disease cases and healthy samples[8,18]. Alternatively, several approaches are based on the topological decomposition of the pathway maps originally proposed in 2005 (ref. 19). These approaches tend to give more weight to the genes that occupy the central positions on the map[20]. Importantly, however, the measure of gene centrality strongly varies among algorithms, often leading to highly variable results.

Here we propose a novel approach that integrates different analytical concepts described above into a single network model as it simultaneously exploits statistical and topological weights for gene importance estimation (Fig. 1). The smooth threshold based on the $P$ values from a $t$-test performed on groups of normal and tumour samples is applied to the gene expression values. The smooth threshold is defined as a continuous function of $P$ value ranging from 0 to 1. The statistical weights for genes are also derived during this procedure. The topological weights for genes are obtained during the pathway map decomposition. The topological weight of each gene is proportional to the number of independent paths through the pathway gene network represented as a directed graph.

It is well known that multiple genes exhibit considerable correlations in their expression levels[21]. Most algorithms for pathway analysis treat gene expression levels as independent variables, which, despite the common belief, is not suitable when the topology based coefficients are applied. Indeed, due to exchangeability, there is no dependence of pathway activation values on how the topology weights are distributed over a set of

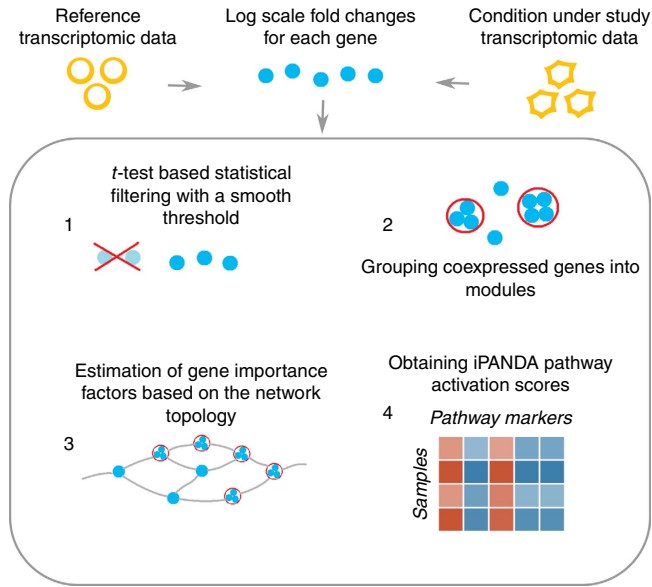

**Figure 1 | The general scheme of iPANDA calculation pipeline.** Fold changes between the gene-expression levels in the samples under investigation, and an average expression level of samples within the normal set serves as input data for the iPANDA algorithm. The major steps of iPANDA algorithm include estimation of statistical weights (**1**), co-expression-based grouping of genes into modules (**2**), estimation of topological weights (**3**) and calculation of iPANDA pathway activation scores (**4**).

coexpressed genes with correlated expression levels, and hence correlated fold changes. Thus, the computation of topological coefficients for a set of coexpressed genes is inefficient, unless a group of coexpressed genes is being considered as a single unit. To circumvent this challenge, gene modules reflecting the coexpression of genes are introduced in the iPANDA algorithm. The wide database of gene coexpression in human samples, COEXPRESdb[21], and the database of the downstream genes controlled by various transcriptional factors[22] are utilized for grouping genes into modules. In this way, the topological coefficients are estimated for each gene module as a whole rather than for individual genes inside the module.

The contribution of gene units (including gene modules and individual genes) to pathway activation is computed as a product of their fold changes in logarithmic scale, topological and statistical weights. Then the contributions are multiplied by a discrete coefficient which equals to $+1$ or $-1$ in the case of pathway activation or suppression by the particular unit, respectively. Finally, the activation scores, which we refer to as iPANDA values, are obtained as a linear combination of the scores calculated for gene units that contribute to the pathway activation/suppression. Therefore, the iPANDA values represent the signed scores showing the intensity and direction of pathway activation (see Methods section for details).

**Pathway quality metrics.** Although currently there are several publicly available pipelines for benchmarking the transcriptomic data analysis algorithms[7,23–25], our aim was to generalize the approaches for pathway-based algorithm testing and reveal the common features of reliable pathway-based expression data analysis. We term these features 'pathway analysis quality hallmarks'. Efficient methods for pathway-based transcriptomic data analysis should be capable to perform a significant noise reduction in the input data and aggregate output data as a small number of highly informative features (pathway markers).

Scalability (the ability to process pathways with small or large numbers of genes similarly) is another critical aspect that should be considered when designing a reliable pathway analysis approach, since pathway activation values for pathways of different sizes should be equally credible. The list of pathway markers identified should be relevant to the specific phenotype or medical condition, and robust over multiple data sets related to the process or biological state under investigation. The calculation time should be reasonable to allow high-throughput screening of large transcriptomic data sets. To address the iPANDA algorithm in respect to these hallmarks and to fully assess its true potential and limitations, we have directly compared the results obtained by iPANDA using the breast cancer and Microarray Analysis Quality Control (MAQC)-I data sets with five other widely used third-party viable alternatives (GSEA[8], SPIA[9], Pathway Level Analysis of Gene Expression (PLAGE)[26], single sample Gene Set Enrichment Analysis (ssGSEA)[27] and Denoising Algorithm based on Relevant network Topology (DART)[28]). Moreover, we compared the performance of our iPANDA-based classifier to performance of the gene-level predictors developed by the best MAQC-II (ref. 23) and IMPROVER[24] teams using the data sets from MAQC-II challenge in respect to the ability to discriminate between cancer end points.

**iPANDA as a tool for noise reduction in transcriptomic data.** One of the major issues that should be addressed when developing a novel transcriptomic data analysis algorithm is the ability of the proposed method to reduce noise while retaining the biologically relevant information of the results. Since pathway-based analysis algorithms are considered dimension reduction techniques, the pathway activation scores should represent collective variables describing only biologically significant changes in the gene expression profile.

In order to estimate the ability of the iPANDA algorithm to perform noise reduction while preserving biologically relevant features, we performed an analysis of the well-known MAQC data set (GEO identifier GSE5350) (ref. 29). It contains data for the same cell samples processed using various transcriptome profiling platforms. A satisfactory pathway or network analysis algorithm should reduce the noise level and demonstrate a higher degree of similarity between the samples in comparison to the similarity calculated using gene set data. To estimate gene level similarity only fold changes for differentially expressed genes ($t$-test $P$ value $<0.05$) were utilized. Pearson correlation was chosen as a metric to measure the similarity between samples. Sample-wise correlation coefficients were obtained for the same samples profiled on Affymetrix and Agilent platforms. Similar procedure was performed using pathway activation values (iPANDA values). The results acquired for the set of samples from MAQC data are shown in Fig. 2. Notably, the similarity calculated using pathway activation values generated by the iPANDA algorithm significantly exceeds the one calculated using fold changes for the differentially expressed genes (mean sample-wise correlation was over 0.88 and 0.79, respectively). To further validate our algorithm, we directly compared its noise reduction efficacy with that of other routinely used methods for transcriptome-based pathway analysis, such as SPIA, GSEA, ssGSEA, PLAGE and DART (Supplementary Fig. 1). The mean sample-wise correlation between platforms was 0.88 for iPANDA compared with 0.53 for GSEA, 0.84 for SPIA, 0.69 for ssGSEA, 0.67 for PLAGE and 0.41 for DART. Furthermore, the sample-wise correlation distribution obtained using iPANDA values is narrowed to a range of 0.79 to 0.94, compared with $-0.08$–$0.80$, 0.60–0.92, 0.61–0.74, 0.45–0.75 and $-0.11$–$0.60$ for GSEA, SPIA, ssGSEA, PLAGE and DART, respectively (Supplementary Fig. 1).

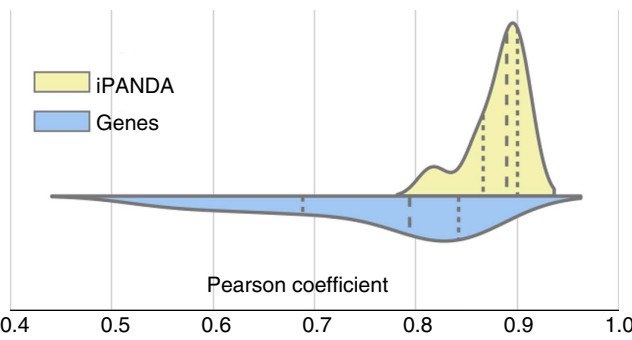

**Figure 2 | Sample-wise similarity between data obtained using various profiling platforms.** Pearson sample-wise correlation coefficients between gene expression levels (differential genes only are used with group *t*-test *P* value <0.05) obtained with Affymetrix and Agilent platforms for the same set of samples are shown in blue. Pearson sample-wise correlations between corresponding pathway activation values calculated using iPANDA are shown in yellow. Dashed and dotted lines represent, respectively, the median with upper and lower quartiles of the empirical distribution. Gene expression data was obtained from MicroArray Quality Control (MAQC) data set (GEO identifier GSE5350). Application of iPANDA leads to higher correlation between the data obtained using different experimental platforms for the same samples.

It is important to mention that iPANDA does not assign more weights to genes that tend to be reliably coexpressed using information from COEXPRESSdb database. The information from COEXPRESSdb is utilized solely for grouping genes into modules, and hence cannot introduce any favourable bias towards iPANDA in this assessment. Even when the feature for grouping genes into modules is 'switched off', meaning that all genes are considered individually and no information from COEXPRESSdb is being utilized, iPANDA scores show higher sample-wise similarity between data obtained using various profiling platforms compared with the similarity calculated on the gene level (Supplementary Fig. 2).

Taken together, iPANDA demonstrates better performance for the noise reduction test in comparison to other pathway analysis approaches, suggesting its credibility as a powerful tool for noise reduction in transcriptomic data analysis.

**Biomarker identification and relevance.** As a next step we have addressed the iPANDA ability to identify potential biomarkers (or pathway markers) of the phenotype under investigation. One of the commonly used methods to assess the capability of transcriptomic pathway markers to distinguish between two groups of samples (for example, resistance and sensitivity to treatment) is to measure their receiver operating characteristics area under curve (AUC) values. The capacity to generate a high number of biomarkers with high AUC values is a major requirement for any prospective transcriptomic data analysis algorithm to be used in prediction models.

To estimate the capability of our method to produce potential biomarkers, we have selected several gene expression data sets from breast cancer patients with measured response to paclitaxel treatment. iPANDA algorithm was applied to obtain pathway activation scores for each sample. For each breast cancer data set used in this study, we have carefully selected a tissue specific normal control (microarrays derived from the healthy subjects using the same profiling platform as in tumour data set, see Supplementary Table 2). *t*-test *P* values for genes were calculated over the whole group of breast cancer samples against healthy tissue samples in order to estimate the statistical weights, which

were further used to obtain sample-wise pathway activation iPANDA scores. Cross-validation approach using samples from GSE20194 data set was utilized to obtain the threshold values for calculation of statistical weights and merging the gene modules. To avoid introduction of the artificial inter-data sets bias for the gene weights as a result of this approach, the gene weights in iPANDA were calculated for each of the data sets used for this analysis separately (independently).

Lists of the top 30 paclitaxel treatment sensitivity pathway markers obtained for the estrogen receptor negative (ERN) HER2-positive (HER2P) and ERN HER2-negative (HER2N) breast cancer types are given in Fig. 3. Four and five independent data sets were used for comparison of ERN HER2P and ERN HER2N cancer types, respectively. Signalling pathways were ranked by their average AUC values over independent data sets examined. Pathways like ERBB, PTEN, BRCA1, PPAR, TGF-beta and RAS, previously reported to trigger paclitaxel treatment response, can be found in these lists[30–34]. Although the iPANDA-generated pathway marker lists obtained within data on the same cancer type have noticeable intersection, the lists of markers differ significantly between cancer types. This complies with the observation that the mechanisms of paclitaxel treatment resistance depend on the breast cancer subtype[35,36].

Pathways with various numbers of member genes ranging from <10 members (vascular endothelial growth factor pathway adhesion turnover) to over 400 (AKT Signalling Main pathway) can be found in the lists. This illustrates that iPANDA algorithm treats small and large pathways in the same way, indicating the scalability hallmark of valid pathway analysis methods.

Similar calculations were performed by using five different third-party pathway analysis algorithms such as GSEA, SPIA, PLAGE, DART and ssGSEA. As demonstrated in Supplementary Figs 3–7, the number of robust pathway markers and corresponding AUC values for these markers obtained by each one of the third-party methods was substantially lower compared with iPANDA.

To further estimate the ability of iPANDA to detect relevant pathways, we have assessed its performance by using the prioritization criteria according to recently proposed pathway methods benchmarking pipeline[25]. In this pipeline, prioritization represents the ability of the method to assign higher ranks to pathways relevant to a given condition in a test with direct comparison between two groups of samples. Although the iPANDA algorithm did not surpass the alternative methods which were reported to be the best according to the prioritization criteria (PADOG[37] and MIPA[38]) (Supplementary Fig. 8 and Supplementary Note 1), it outperforms some other popular methods including ssGSEA and PLAGE, and demonstrates the ability to generate highly relevant results, since the pathways expected to be perturbed have significantly lower ranks (higher scores) than if it was expected by chance. Moreover, the prioritization pipeline relies on a very specific set of pathways expected to be perturbed under certain disease conditions. Each of these pathways consists of genes associated with multiple mechanisms of biological regulation, therefore, these pathways contain several sparsely interconnected components. In contrast, iPANDA is specifically designed to address regulatory circuits with well-defined topology (for example, mTOR pathway, AKT pathway, and so on.). Hence, the design of this particular prioritization assessment may subsequently lead to an underestimation of iPANDA performance.

**iPANDA produces highly robust set of biomarkers.** One of the most important shortcomings of modern pathway analysis approaches is their inability to produce consistent results for different data sets obtained independently for the same biological

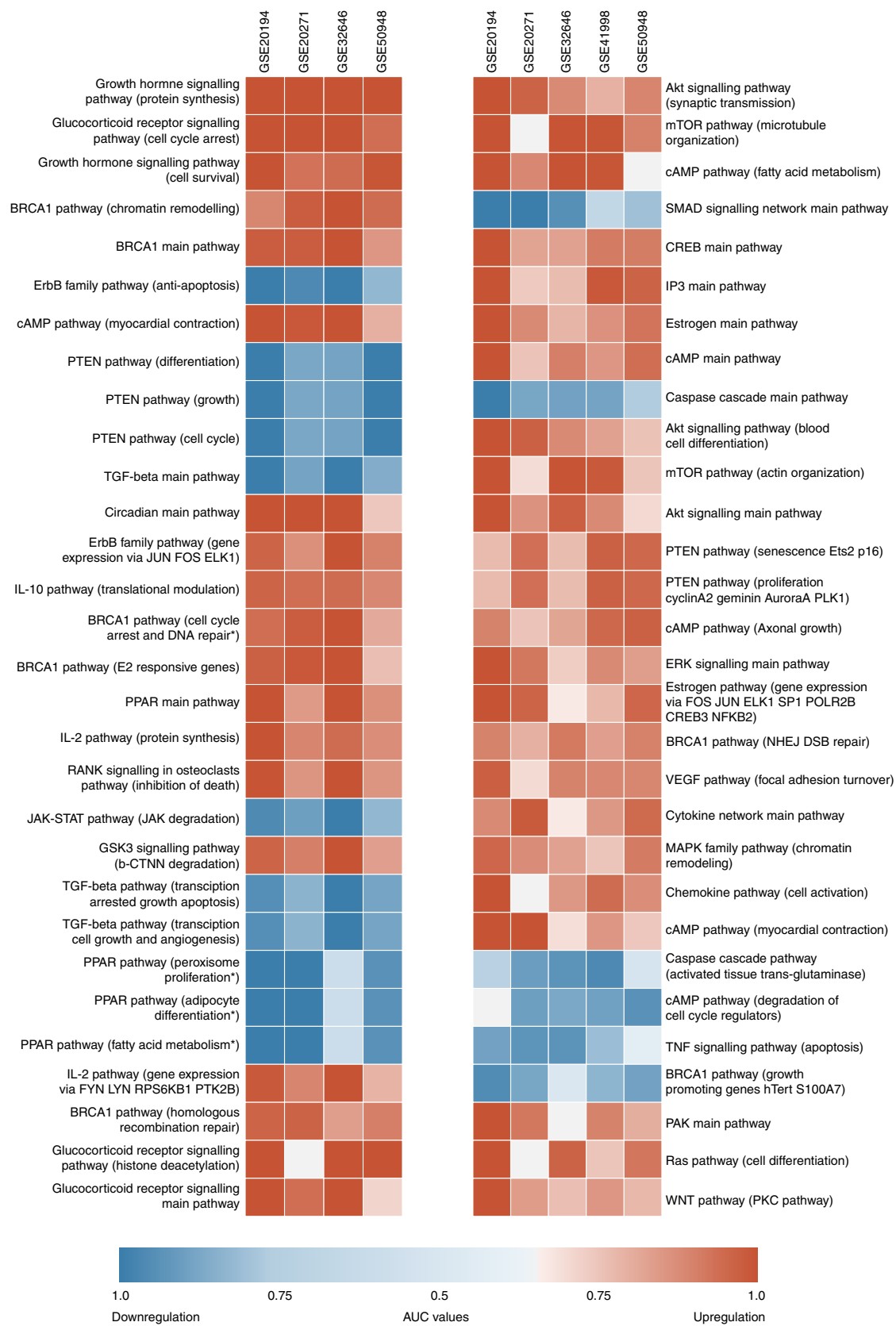

**Figure 3 | Receiver operating characteristic AUC values for 30 highest rated by AUC pathway markers.** Pathway markers of responders/non-responders to paclitaxel for ERN HER2P (left) and ERN HER2N (right) breast cancer treatment were obtained using iPANDA. Up and downregulated pathways in responders group compared with non-responders group are shown in red and blue, respectively. The saturation of the colour denotes to corresponding AUC value. The same signalling pathways are found to be markers of responders/non-responders to paclitaxel treatment for four (ERN HER2P) or five (ERN HER2N) independent data sets obtained from GEO. Nineteen and eight pathway markers for ERN HER2P and ERN HER2N breast cancer, respectively, demonstrate AUC values higher than 0.7 for all data sets examined.

case. Here we show that iPANDA algorithm applied to the breast cancer data overcomes this flaw and produces highly consistent set of pathway markers across the data sets used in the study. In particular, the iPANDA values for 19 and 8 pathways for ERN HER2P and ERN HER2N breast cancer types, respectively, can be utilized as paclitaxel response classifiers with AUC values higher than 0.7 for all data sets examined. Whereas, all third-party algorithms tested (including GSEA, SPIA, DART, ssGSEA and PLAGE) failed to obtain even a single pathway marker (with the AUC threshold equal to 0.7) common for all data sets examined (for both cancer types) (Supplementary Figs 3–7), suggesting that iPANDA algorithm is an advantageous method for biologically relevant pathway marker development compared with the other pathway-based approaches.

The common marker pathway (CMP) index (see Methods section for details) was applied to paclitaxel treatment response data for both ERN HER2P and ERN HER2N breast cancer types in order to estimate the robustness of the biomarker lists. Pathway marker lists obtained for four independent data sets were analysed. The calculation of pathway activation scores was performed using the iPANDA algorithm and its versions with disabled gene grouping and/or topological weights (Fig. 4). The 'off' state of topology coefficients means that they are equal to 1 for all genes during the calculation. Also, the 'off' state for the gene grouping means that all the genes are treated as individual genes. The application of the gene modules without topology-based coefficients reduces the robustness of the algorithm as well as the overall number of common pathway markers between data sets. Turning on the topology-based coefficients just slightly increases the robustness of the algorithm. Whereas using topology and gene modules simultaneously dramatically improves this parameter for both cancer types. This result implies that the combined implementation of the gene modules along with the topology-based coefficients serves as an effective way of noise reduction in gene expression data and allows one to obtain stable pathway activation scores for a set of independent data.

**iPANDA biomarkers as classifiers for prediction models**. High AUC values for the pathway markers shown in Fig. 3 suggest that iPANDA scores may be efficiently used as classifiers for biological condition prediction challenges. To test this hypothesis, pathway activation scores obtained using iPANDA were applied to the identification of paclitaxel neoadjuvant therapy sensitivity in

breast cancer. The normalized iPANDA scores were calculated for all samples in six data sets and merged for the paclitaxel treatment response prediction procedure (Table 1). Prediction models were built for three separate end points: paclitaxel sensitivity in ERN HER2N tumours only, in ERN HER2P tumours only and in all breast cancer types merged together (including those with ERP HER2N type and tumours with unknown receptor status). Samples were divided into training and validation sets to measure the performance of prediction models in respect to the end points under study on the data set-wise basis. Samples from GSE20194 and GSE20271 data sets were used as training, while samples from other data sets, where samples with particular cancer type under consideration were available, were used for validation. Not all of the six data sets used for the analysis contain samples with both cancer types. Statistical weights were obtained using the whole group of case samples (including both responders and non-responders) and the group of paired normal samples (Supplementary Table 2) separately for training and validation sets. The patients' clinical outcome used for benchmarking (response to paclitaxel) was not exposed to iPANDA in any way, which precludes information leakage across different phenotypes.

In order to classify the samples as responders or non-responders, the random forest models were developed using iPANDA scores obtained for training sets of samples for each end point (see Methods section). Subsequently, performance of these models was measured using validation sets. Matthew's Correlation Coefficients (MCC), specificity and sensitivity metrics were applied to evaluate performance of the models (Fig. 5). MCC metrics were chosen for the ease to calculate and due to their informativeness even when the distribution of the two classes is highly skewed[23]. The similar random forest models were built using pathway activation (enrichment) scores obtained by other pathway analysis algorithms, including SPIA, GSEA, DART, ssGSEA and PLAGE. Moreover, to fully assess the performance of iPANDA-based paclitaxel sensitivity prediction models, we have trained the similar random forest models on four different gene expression subsets: expression levels of all genes (logGE), fold change for all genes between the training set and corresponding normals (logFC), expression levels of most differentially expressed genes ($t$-test $P < 0.05$) (logDGE), and fold change in expression levels of most differentially expressed genes ($t$-test $P < 0.05$) between the training and corresponding normal breast tissue data sets (logDFC). Logarithmic scale was used for training the gene level models. All pathway-level and gene-level data was Z-score normalized separately for each GEO data set used (see online Methods for details).

As demonstrated in Fig. 5, the models developed using normalized iPANDA scores distinguished paclitaxel treatment responders from non-responders with high accuracy. Furthermore, after Z-score normalization of the iPANDA scores, high accuracy was achieved for all the data sets used for validation, regardless of the differences across these data sets (Supplementary Data 1). The MCC for iPANDA-based paclitaxel response prediction model in ERN HER2P patients equals to 0.758 with specificity and sensitivity of 0.949 and 0.778, respectively. ERN HER2N breast cancer and especially its triple negative subclass (also progesterone receptor negative) is known to have the most diverse phenotype[39]. Therefore, prediction of the therapy outcome for this type of breast cancer is a challenging task. Nevertheless, the application of iPANDA values as input for random forest classifiers for paclitaxel treatment response prediction in ERN HER2N breast cancer shows relatively high accuracy, with MCC, specificity and sensitivity of the model equal to 0.532, 0.930 and 0.545, respectively. While this result is lower compared with prediction accuracy obtained for ERN HER2P

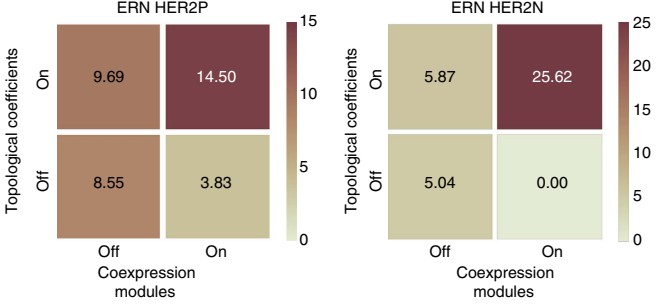

**Figure 4 | Common marker pathway (CMP) index for responders/ non-responders to paclitaxel treatment of ERN HER2P and ERN HER2N breast cancer types.** The index is calculated for four independent data sets obtained from GEO (GSE20194, GSE20271, GSE32646 and GSE50948) for each cancer type. Index demonstrates the robustness of the pathway marker lists between data sets. Independent application of the gene modules and topological coefficients did not improve the robustness of the algorithm for estimation of pathway activation; however, combined application resulted in significant improvement.

**Table 1 | Training and validation data sets used in paclitaxel neoadjuvant therapy sensitivity prediction experiment in breast cancer patients.**

| End point | Training set | | Validation set | |
|---|---|---|---|---|
| | GEO data sets | Number of samples | GEO data sets | Number of samples |
| ERN HER2P | GSE20194, GSE20271 | 38 | GSE32636, GSE50948 | 57 |
| ERN HER2N | GSE20194, GSE20271 | 108 | GSE32646, GSE41998, GSE50948 | 115 |
| All cancer types | GSE20194, GSE20271 | 285 | GSE22513, GSE32646, GSE41998, GSE50948 | 299 |

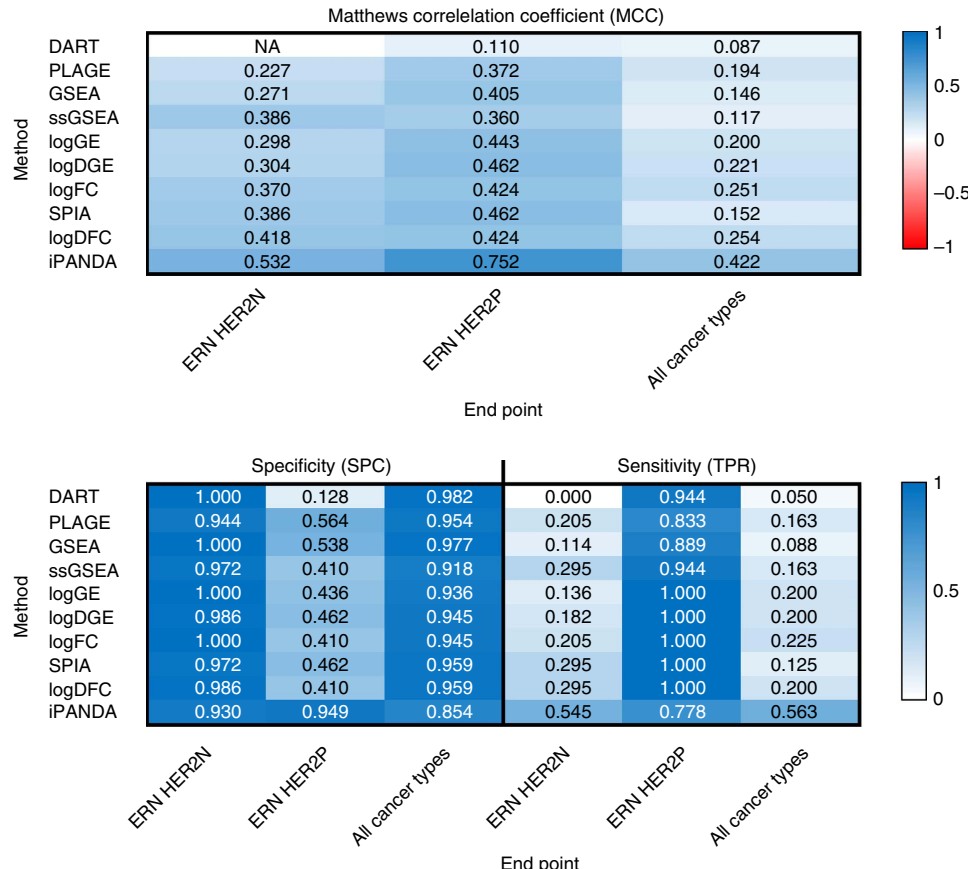

**Figure 5 | Performance of random forest models for paclitaxel sensitivity prediction in breast cancer patients.** Models were built in respect to three distinct end points: sensitivity of ERN HER2P cancer type, ERN HER2N type and all breast cancer types mixed. The models were trained using iPANDA, SPIA, GSEA, ssGSEA, PLAGE and DART pathway activation (enrichment) scores and gene-level data including: gene expression values for all genes (logGE), fold changes of tumour samples relative to the mean of paired normal samples for all genes (logFC), gene expression for differential genes only (genes are considered differentially expressed if $t$-test $P$ value was $< 0.05$ between tumour and normal samples, logDGE) and fold changes for differential genes only (logDFC). MCC, specificity and sensitivity performance metrics are shown for each model.

cancers, it is sufficient for considering further evaluation of the proposed model as a future decision making support system in clinic.

Moreover, taxol-based neoadjuvant therapy response prediction with iPANDA-based random forest model was performed using mixed data regardless of the cancer type of the samples. The performance metrics of the model were 0.422, 0.854 and 0.563 for MCC, specificity and sensitivity, respectively. Notably, the performance of the model on samples from various data sets used for validation was not homogeneous. Interestingly, all 28 samples (8 responders and 20 non-responders) from the GSE22513 data set were correctly separated into two groups (MCC equals 1.000), whereas for the GSE41998 data set the MCC value generated by the model was only 0.254. This observation demonstrates that the distinctions between data sets can be blurred, but not completely eliminated when developing

prediction models. Nevertheless, the results show that patients' classification according to their potential treatment response can be successful even in case when receptor status is unknown. These finding suggest that despite the differences between breast cancer types, both ERN HER2P and ERN HER2N cancer share common features which can define treatment response sensitivity.

Although we acknowledge that the methods used for comparison in this study were initially designed for the relevant pathway assessment for a given condition rather than for phenotype prediction, we have selected these methods for comparison as they are among the most cited and highly acclaimed algorithms in the community. Nevertheless, we expect that a good pathway analysis approach would produce robust scores, which should demonstrate a certain degree of discriminative power if the pathway database is chosen accordingly to the biological condition under study. Comparison to other models

based on pathway activation (enrichment) scores obtained with other pathway activation algorithms shows that iPANDA-based classifiers outperform these methods for all three end points examined. Moreover, iPANDA surpasses the similar random forest models trained on gene-level data, implying that iPANDA is an effective tool for noise reduction in gene expression data, while preserving the biologically relevant features. Interestingly, using fold changes between tumour samples and normal samples instead of solely gene expression increases the predictive power of the model, while the use of differentially expressed genes only does not lead to significant improvement. Moreover, pathway analysis algorithms that explicitly account for fold changes (iPANDA and SPIA) generate better results when used as inputs for prediction models in comparison to other methods. This observation demonstrates that direct incorporation of the expression data from corresponding normal tissue into the fold change calculation, can be a valuable addition when developing prediction models.

To further evaluate the proficiency of iPANDA for reliably classifying a sample with respect to the various clinical parameters, we have used the training and validation data sets (refer to Supplementary Note 2 and Supplementary Fig. 9) along with the best gene-level predictive models reported by the MAQC-II project as a reference for comparison. We have also compared the prediction power of our iPANDA-based random forest classifier to the best methods from the IMPROVER challenge. Using the iPANDA scores for cancer-related pathways to train prediction models, allows to obtain significantly better results compared with the gene-level prediction models developed by either our team or MAQC-II consortium (for 3 out of 5 cancer end points available for comparison F, J, K). The highest increase in performance was achieved on neuroblastoma event free survival (end point K) (0.894 against 0.575 for the best MAQC-II team). These observations further support our notion that iPANDA algorithm can provide an efficient noise reduction when extracting biologically relevant features from the data, compared with other methods. Hence iPANDA may be a useful tool, when used as input for machine learning algorithms to make better prediction models.

## Discussion

Application of the pathway activation measurement implemented in iPANDA leads to significant noise reduction in the input data and hence enhances the ability to produce highly consistent sets of biologically relevant biomarkers acquired on multiple transcriptomic data sets. Another advantage of the approach presented is the high speed of the computation. The gene grouping and topological weights are the most demanding parts of the algorithm from the perspective of computational resources. Luckily, these steps can be precalculated only once before the actual calculations using transcriptomic data. The calculation time for a single sample processing equals ∼1.4 s on the Intel (R) Core i3-3217U 1.8 GHz CPU (compared with 10 min for SPIA, 4 min for DART, about 10 s for ssGSEA, GSEA and PLAGE). Thus, iPANDA can be an efficient tool for high-throughput biomarker screening of large transcriptomic data sets.

The use of merely microarray data for pathway activation analysis has well-known limitations, as it cannot address individual variations in the gene sequence and consequently in the activity of its product. For example, a gene can have a mutation that reduces activity of its product but elevates its expression level through a negative feedback loop. Thus, the elevated expression of the gene does not necessarily corresponds with the increase in the activity of its product. Nevertheless, comprehensive analysis of the tumour pathway activation profile may be a more clinically relevant strategy to stratify the subset of

patients whose tumours could probably respond and who would clinically benefit from anti-cancer therapeutic regimens than other outcome prediction methods based on the gene expression profile. While gene expression levels can be effectively used for phenotype prediction, it is quite possible that the most differentially expressed genes in a given signature will not be part of the pathways that actually drive tumour behaviour. Alternatively, expression of some genes within the cancer driving pathways is not always predictive of the overall pathway activation. Therefore, while there is no single preferential approach for interpreting gene expression results, the proposed method of transcriptomic data analysis on the signalling pathway level may not only be useful for discrimination between various biological or clinical conditions, but may aid in identifying functional categories or pathways that may be relevant as possible therapeutic targets.

Although the iPANDA algorithm was initially designed for microarray data analysis, it can also be easily applied to the data derived from genome-wide association studies (GWAS). In order to do so, GWAS data can be converted to a form amenable for the iPANDA algorithm. Single-point mutations are assigned to the genes based on their proximity to the reading frames. Then each single-point mutation is given a weight derived from a GWAS data statistical analysis[40]. Simultaneous use of the GWAS data along with microarray data may improve the predictions made by the iPANDA method.

One of the rapidly emerging areas in biomedical data analysis is deep learning[41]. Recently several successful studies on microarray data analysis using various deep learning approaches on gene-level data have surfaced[42]. Using pathway activation scores may be an efficient way to reduce dimensionality of transcriptomic data for drug discovery applications while maintaining biological relevant features[43]. From an experimental point of view, gene regulatory networks are controlled via activation or inhibition of a specific set of signalling pathways. Thus, using the iPANDA signalling pathway activation scores as input for deep learning methods could bring results closer to experimental settings and make them more interpretable to bench biologists. One of the most difficult steps of multilayer perceptron training is the dimension reduction and feature selection procedures, which aim to generate the appropriate input for further learning[44]. Signalling pathway activation scoring using iPANDA will likely help reduce the dimensionality of expression data without losing biological relevance and may be used as an input to deep learning methods especially for drug discovery applications. Using iPANDA values as an input data seems to be a particularly promising approach to obtaining reproducible results when analysing transcriptomic data from multiple sources.

## Methods

**Transcriptomic data.** From the GEO database we have downloaded six data sets containing gene expression data related to breast cancer patients treated with paclitaxel and three data sets containing transcriptomic data from normal cancer-free breast tissue which were used as a reference (Supplementary Table 2).

The breast cancer and normal data from different data sets was preprocessed using GCRMA algorithm[45] and summarized using updated chip definition files from Brainarray repository (Version 18) (ref. 46) for each data set independently.

ERN breast cancer samples were stratified by the HER2 status: HER2-positive (HER2P) and HER2-negative (HER2N). Only samples profiled before any treatment were analysed. In this analysis, we included the samples from patients who subsequently underwent paclitaxel (Taxol) treatment and showed any definite therapy outcome (response or non-response) (Supplementary Table 3). Also, GSE22513 breast cancer data set with known paclitaxel treatment outcome and unspecified receptor status was utilized. It contains samples from 20 non-responders and 8 responders patients.

The MicroArray Quality Control (MAQC) data set (GEO identifier 5350) was obtained from the GEO database. The raw data for 60 samples from Affymetrix platform was preprocessed using GCRMA algorithm[45] and summarized using updated chip definition files from Brainarray repository (Version 18) (ref. 46) for

each data set independently. The preprocessed data for 60 samples from Agilent platform was taken as provided by authors. These samples represent four different groups: A = Stratagene Universal Human Reference RNA (UHRR, Catalog #740000), Sample B = Ambion Human Brain Reference RNA (HBRR, Catalog #6050), Sample C = Samples A and B mixed at 75%:25% ratio (A:B); and Sample D = Samples A and B mixed at 25%:75% ratio (A:B). Group A was used as a reference.

**Pathway database overview.** There are several widely used collections of signalling pathways including Kyoto Encyclopedia of Genes and Genomes, QIA-GEN and NCI Pathway Interaction Database. In this study, we use the collection of signalling pathways most strongly associated with various types of malignant transformation in human cells obtained from the SABiosciences collection (http://www.sabiosciences.com/pathwaycentral.php). Using a cancer-specific pathway database ensures the presence of multiple pathway markers for the particular phenotype of the breast cancer under investigation. The database contains a set of 374 signalling pathways which cover a total of 2,294 unique genes. Each pathway contains an explicitly defined topology represented as a directed graph. Each node corresponds to a gene or a set of genes while edges describe biochemical interactions between genes in nodes and/or their products. All interactions are classified as activation or inhibition of downstream nodes. The pathway size ranges from about twenty to over six hundred genes in a single pathway.

**Estimation of pathway activation.** Our novel approach for large-scale transcriptomic data analysis accounts for the gene grouping into modules based on the precalculated gene coexpression data. Each gene module represents a set of genes which experience significant coordination in their expression levels and/or are regulated by the same expression factors (see grouping genes section below). Therefore, the actual function for the calculation of the pathway $p$ activation according to the proposed iPANDA algorithm consists of two terms. While the first one corresponds to the contribution of the individual genes, which are not members of any module, the second one takes into account the contribution of the gene modules. Therefore, the final function for obtaining an iPANDA value for the activation of pathway $p$, which consists of the individual genes $i$ and gene modules $j$, has the following analytical form:

$$\text{iPANDA}_p = \sum_i G_{ip} + \sum_j M_{jp} \qquad (1)$$

The contribution of the individual genes ($G_{ip}$) and the gene modules ($M_{jp}$) is computed as follows:

$$G_{ip} = w_i^S \cdot w_{ip}^T \cdot A_{ip} \cdot \log(fc_i) \qquad (2)$$

$$M_{jp} = \max(w_i^S) \cdot \frac{1}{N} \sum_i^N \left( w_{ip}^T \cdot A_{ip} \cdot \log(fc_i) \right) \qquad (3)$$

Here $fc_i$ is the fold change of the expression level for the gene $i$ in the sample under study to the normal level (average in a control group). As the expression levels are assumed to be logarithmically normally distributed and in order to convert the product over fold change values to sum, logarithmic fold changes are utilized in the final equation. Activation sign $A_{ip}$ is a discrete coefficient showing the direction in which the particular gene affects the pathway given. It equals $+1$ if product of the gene $i$ has a positive contribution to the pathway activation and $-1$ if it has a negative contribution. The factors $w_i^S$ and $w_{ip}^T$ are the statistical and topological weights of the gene $i$ ranging from 0 to 1. The derivation procedure for these factors is described in detail in the subsequent sections. Since $\log(fc_i)$ and $A_{ip}$ values can be positive or negative, the iPANDA values for the pathways can also have different signs. Thus, positive or negative iPANDA values correspond to pathway activation or inhibition respectively. The principal scheme of the iPANDA algorithm is shown in Fig. 1.

**Obtaining gene importance factors.** In order to estimate the topological weight ($w_{ip}^T$), all possible walks through the gene network are calculated on the directed graph associated with the pathway map. The nodes of the graph represent genes or gene modules, while the edges correspond to biochemical interactions. The nodes which have zero incoming edges are chosen as the starting points of the walks and those which have zero outgoing edges are chosen as the final points. Loops are forbidden during walks computation. The number of walks $N_{ip}$ through the pathway $p$, which include gene $i$ is calculated for each gene. Then $w_{ip}^T$ is obtained as the ratio of $N_{ip}$ to the maximum value of $N_{jp}$ over all gene unit in the pathway:

$$w_{ip}^T = \frac{N_{ip}}{\max(N_{jp})} \qquad (4)$$

The statistical weight depends on the $P$ values, which are calculated from $t$-test for case and normal sets of samples for each gene. The method called $P$ value thresholding is commonly used to filter out spurious genes, which demonstrate no significant differences between sets. However, a major issue with the use of sharp threshold functions is that it can introduce an instability in filtered gene sets and as

a consequence in pathway activation scores between the data sets. Additionally, the pathway activation values become sensitive to an arbitrary choice of the cutoff value. In order to address this issue, we suggest using a smooth threshold function. In the present study, the cosine function on logarithmic scale is utilized:

$$w_i^S = \begin{cases} 0, p > p_{\max} \\ \left( \cos\left( \pi \frac{\log p - \log p_{\min}}{\log p_{\max} - \log p_{\min}} \right) + 1 \right) \Big/ 2, p_{\min} < p \le p_{\max} \\ 1, p \le p_{\min} \end{cases} \qquad (5)$$

where $p_{\min}$ and $p_{\max}$ are the low- and high-threshold values. In this study, $P$ value thresholds equal to $10^{-7}$ and $10^{-1}$, respectively. For the threshold values given over 58% of all genes pass high threshold and about 12% also pass low threshold for the breast cancer data under investigation. Hence, over 45% of the genes in the data set receive intermediate $w_i^S$ values. Therefore more stable results for pathway activation scores between data sets can be achieved using this approach.

**Grouping genes into modules.** To obtain the gene modules, two independent sources of data were utilized: human database of coexpressed genes COEXPRESdb[21] and the database of the downstream genes controlled by human sequence-specific transcription factors[22]. The latter was simply intersected with the genes from the pathway database used, while correlation data from COEXPRESdb was clustered using Euclidean distance matrix. Distances were obtained according to the following equation:

$$r_{ij} = 1 - \text{corr}_{ij} \qquad (6)$$

where $\text{corr}_{ij}$ is correlation between expression levels of genes $i$ and $j$. DBScan and hierarchical clustering with an average linkage criteria were utilized to identify clusters. Only clusters with an average internal pairwise correlation higher than 0.3 were considered. Clusters obtained from the transcription factors database and coexpression database were recursively merged to remove duplicates. A pair of clusters was combined into one during the merging procedure if the intersection level between clusters had been higher than 0.7. As a result, a set of 169 gene modules which includes a total of 1,021 unique genes was constructed.

**Statistical credibility of the iPANDA values.** The $P$ values for the iPANDA pathway activation scores are obtained using weighted Fisher's combined probability test. Thus the $P$ values ($p_p$) are estimated according to the following equation:

$$\ln(p_p) = \frac{\sum_i^N \left( w_i^S \cdot w_{ip}^T \cdot \ln(p_i) \right)}{\sqrt{\sum_i^N \left( w_i^S \cdot w_{ip}^T \right)^2}} \qquad (7)$$

where $i$ refers to the particular member (individual gene or gene module) of the pathway $p$, $N$ is the number of pathway members, $w_i^S$ and $w_{ip}^T$ are the topological and statistical weights of the member $i$, $p_i$ is the group $t$-test $P$ value for the member $i$. Since the $P$ values obtained are not used for further pathway marker scoring they are not normalized in any way and rely solely on the statistics for the genes which have non-zero statistical weights.

**Algorithm robustness estimation.** In order to quantitatively estimate the robustness of the algorithm between data sets, we introduce the CMP index. The CMP index is a function of the number of pathways considered as markers that are common between data sets. It also depends on the quality of the treatment response prediction when these pathways are used as classifiers. The CMP index is defined as follows:

$$\text{CMP} = \frac{1}{n} \sum_{j=1}^n \sum_i \ln(N_i) \times (\text{AUC}_{ij} - \text{AUC}_R) \qquad (8)$$

where $n$ is the number of data sets under study, $N_i$ is the number of genes in the pathway $i$ and $\text{AUC}_{ij}$ is the value of the receiver operating characteristic area under curve, which shows the quality of the separation between responders and non-responders to treatment when pathway $i$ is used as classifier for the $j$-th data set. $\text{AUC}_R$ is the AUC value for a random classifier and equals to 0.5. A pathway is considered as a marker if its AUC value is higher than 0.75. The $\ln(N_i)$ term is included to increase the contribution of the larger pathways because they have a smaller probability to randomly get a high AUC value. The larger values of the CMP index correspond to the most robust prediction of pathway markers across the data sets under investigation, while zero value of CMP index corresponds to the empty intersection of the pathway marker lists obtained for the different data sets.

**Prediction model development.** In order to apply iPANDA to the paclitaxel treatment response prediction over a several independent data sets, the pathway activation values were normalized to the Z-scores independently for each data set. The expected values used for the Z-scoring procedure were adjusted to the number of responders and non-responders in the data set under study. For each end point

described in the result section, random forest classifiers were trained using pathway-activation scores or gene-level data for training set. Then the predictive power of the models was evaluated using validation set. RandomForest function from bioconductoR repository (version 4.6–12) with default settings was utilized to train and run the models (https://cran.r-project.org/web/packages/randomForest/index.html).

To assess the quality of prediction models several metrics including TP—the number of responders samples identified as responders, FP—the number of responders identified as non-responders, FN—the number of non-responders identified as responders, TN—the number of non-responders identified as non-responders, MCC—Matthews correlation coefficient, ACC—accuracy, TRP—sensitivity and SPC—specificity were utilized.

**Performing calculations using third party algorithms.** In order to assess the results obtained with iPANDA algorithm from the perspective of modern advances in the pathway-based transcriptomic data analysis widely used third-party packages were selected for the comparison. The GSEA, single sample version of GSEA (ssGSEA) and SPIA packages were chosen as the most commonly used. PLAGE was selected as the best performing method in the recently proposed pathway analysis method benchmarking approach[25]. DART was reported to perform well specifically on breast cancer data and hence was chosen for the analysis.

We used java GSEA package downloaded from the GSEA official web site (www.broadinstitute.org/gsea/index.jsp). All the input data files were prepared according to the GSEA User guide available at http://software.broadinstitute.org/gsea/doc/GSEAUserGuideFrame.html. Only expression levels for differentially expressed genes (group $t$-test $P$ value $< 0.05$) were used as the input. The pathway database was converted to the GSEA file format using the same package. All the calculations were run from the command line for each tumour sample versus all available normal samples for the particular data set. The parameter 'Number of permutations' was set to 1,000, 'Collapse data set to gene symbols' was set to 'false', 'Permutation type' was set to 'gene_set', 'Enrichment statistics' was set to 'weighted', 'Scoring scheme' was set to 'weighted', 'Metric for ranking genes' was set to 'Signal2Noise', 'Gene list sorting mode' was set to 'real', 'Gene list ordering mode' was set to 'descending', 'Collapsing mode for probe sets' was set to 'max_probe', 'Normalization mode' was set to 'meandiv', 'Randomization mode' was set to 'no balance'. Normalized enrichment score values were extracted from GSEA report for further analysis.

SPIA R package was downloaded from Bioconductor Web site according to the instructions on the SPIA Bioconductor page (http://bioconductor.org/packages/release/bioc/html/SPIA.html). The pathway database was converted to the SPIA file format using the same package. Fold changes between each tumour sample and the mean over the whole set of normal samples for the differentially expressed genes (group $t$-test $P$ value $< 0.05$) were used as the input for the calculations. The total net accumulation perturbation (tA) values for each pathway were extracted from SPIA output for further analysis. All the steps of further pathway analysis using GSEA and SPIA algorithms were similar to the ones used for the analysis performed using iPANDA algorithm. The use of tA values for comparison with other methods has certain limitations. tA is only half of the evidence that SPIA considers and is independent of the information about the overall amount of differential expression in the pathway that the other methods use. Besides, tA is based on information derived only from the genes in the pathway that are connected with certain type of relations documented in Kyoto Encyclopedia of Genes and Genomes, while all other methods use information from all genes in the pathway. Nevertheless, SPIA tA values are the only metric besides the iPANDA values, which surpasses gene-level cross-platform correlations between MAQC-I samples and demonstrates better performance when used in phenotype prediction models comparing with gene-level data.

DART R package (version 1.20.0) was downloaded from bioconductoR repository (https://www.bioconductor.org/packages/release/bioc/html/DART.html). From all studied breast cancer samples, we constructed a normalized gene expression data matrix with logarithmic gene expression values. DART was run independently for each signalling pathway. Each pathway was given to DART as a Model Signature—a numeric vector (in our case, having $+1$ and $-1$ values) reflecting if a gene contributes positively or negatively to the pathway. Relevance network was calculated with the default fdr $= 0.000001$ threshold for correlations between gene expressions. According to DART usage guidelines, we evaluated gene network consistency and pruned each network. We next predicted pathway activation scores in every sample for a given pruned gene network (reflecting a single signalling pathway) and gene expression matrix. Remarkably, DART doesnot construct gene networks for the pathways in which no gene pair has significant correlation—that is why for these pathways activation scores were set to 0 in each sample. Calculated activation scores for each pathway and each sample were constructed to a numeric matrix.

ssGSEA is a single sample extension of GSEA that allows one to define an enrichment score of a gene set in each sample within a given data set[27]. PLAGE[26] is a singular value decomposition-based pathway analysis method similar to GSEA. Both of these methods can be used to develop phenotype prediction models based on transcriptomics. ssGSEA and PLAGE were run with default settings from GSVA[18] package (version 1.16.0) in bioconductoR repository. Cancer-specific pathways database was used as gene sets. We used logarithmic rank-normalized

gene expression data of normal and tumour samples as an input to ssGSEA and PLAGE.

**Preprocessing of MAQC-II data sets.** Three data sets of MAQC-II (GSE20194, GSE24080, GSE49710) and normal samples from GSE9574, GSE13591, GSE19422 as corresponding controls obtained using the same profiling platform were downloaded from GEO[13].

Breast cancer and multiple myeloma data with corresponding normal samples were processed using AFFY (version 1.46.1) and FRMA (version 1.20.0) algorithms from bioconductoR repository. Neuroblastoma data and corresponding normal samples were processed using SCAN.UPS (version 2.10.9) from bioconductoR repository. We divided data into training and validation groups according to the original MAQC-II experiment design[47].

iPANDA scores were calculated for all samples from three cancer-related data sets used in MAQC-II study separately for training and validation sets. Statistical weights were calculated for the whole group of tumour samples compared with corresponding normal samples for training and validation groups independently.

In order to build a prediction model for iPANDA scores we used random forest algorithm implemented in randomForest package from bioconductoR repository (version 4.6–12). All the parameters were set as recommended by default. For each phenotype iPANDA scores for MAQC-II training data sets were used as an input to random forest to make a prediction on 10 MAQC-II end points.

**MaPredictDSC (IMPROVER).** MaPredictDSC (version 1.6.0), the best method from IMPROVER DSC challenge[24], was downloaded from bioconductoR repository. Logarithmic gene expression data was used as input for maPredictDSC. LDA and kNN classifier type, $t$-test and moderate $t$-test were applied during maPredictDSC run. $CVP = 2, NF = 4, NR = 1, FCT = 1.0$ were applied during predictDSC run. The best performing model was selected using validation data sets according to AUC performance metrics. AggregateDSC mode was used to combine the prediction of several models (crowd).

**Data availability.** The code of iPANDA package is deposited to github repository (https://github.com/varnivey/ipanda) and is freely available for academic use. Transcriptomic data sets that support the findings of this study are publicly available from the GEO data sets website (http://www.ncbi.nlm.nih.gov/geo/) using corresponding accession numbers.

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

## Acknowledgements

We are grateful to Dr Adi Tarca (Wayne State University, School of Medicine) and Dr Gautam Goel (Center for Computational and Integrative Biology, Harvard Medical School) for their help and support that have enabled this work to be carried out. We would also like to thank Dr Leslie C. Jellen for editing this manuscript for language and style.

## Author contributions

All authors contributed to the development of iPANDA algorithm design. I.O., K.L., A. Artemov and A. Aliper contributed to the implementation of the algorithm. J.V., A.O., I.L., M.W., A.B., C.C., Y.N., N.B., E.K., D.S., M.C. and A.Z. contributed to the development of iPANDA validation pipeline. I.O., K.L., A. Artemov, E.I., Q.V., S.M. and A.Z. contributed to acquisition, data analysis and testing third-party algorithms used for comparison. I.O., K.L. and S.M. developed and tested prediction models. E.I., D.S., I.I., J.V. and I.L. analysed the breast cancer results in respect to their clinical relevance and possible application. I.O., K.L. and A. Artemov prepared the figures and tables for publication. I.O., K.L., E.I., A. Artemov, Q.V. and A.Z. wrote the manuscript. All authors took part in revising and editing the manuscript.

## Additional information

**Competing financial interests**: Ivan V. Ozerov, Ksenia V. Lezhnina, Quentin Vanhaelen, Sergey Medintsev, Alexander Aliper, Artem V. Artemov, Anton Buzdin, Nikolay Borisov and Alex Zhavoronkov are affiliated with Insilico Medicine, Inc., a company engaged in aging research, which uses and provides both paid and free services using a variety of pathway activation scoring algorithms and hence may have competing financial interests.

**How to cite this article**: Ozerov, I. V. *et al. In silico* Pathway Activation Network Decomposition Analysis (iPANDA) as a method for biomarker development. *Nat. Commun.* **7**, 13427 doi: 10.1038/ncomms13427 (2016).

**Publisher's note**: 

