## [Peer Review File · Nature Communications]

Reviewer #1 (Remarks to the Author):

The article presents a new pathway analysis method that allegedly is better at identifying pathways relevant to a given condition while the pathway summary scores can be used to phenotype prediction. The paper is well written and there is a substantial amount of work that was involved in this manuscript. However, there are several aspects that do not allow to evaluate the usefulness of the new approach as follows:

1) Traditionally pathway analysis methods were introduced to aid in interpreting differential gene expression results by identifying predefined functional categories or pathways that are perturbed in a given phenotype based on gene expression levels. Although the authors claim that there is "no standardized pipeline for benchmarking transcriptomic data analysis algorithms", there are ways to assess whether a given pathway analysis method identifies biological pathways that are relevant to a given dataset as described elsewhere under minimal assumptions (PMID:25147207, PMID:24260172). Moreover, methods to develop prediction models from transcriptomics data have also been evaluated including large scale initiatives such as MAQC-II and IMPROVER using independent datasets for evaluation of the predictive performance. The methods used for these two goals (pathway identification and prediction model development) are different, while the current paper presents iPANDA which is compared to existing methods that were designed for the first goal rather than the latter. So structurally the approach to benchmark the new method is not appropriate.

2) Although this reviewer is open to the idea of using pathway activation scores to improve outcome prediction performance, state of the art methods described in the MAQC-II and IMPROVER challenges, based on gene level predictors were not used as reference, and hence there is no reason why someone would believe that the new approach is worthwhile using. Although the authors present prediction performance results when training and testing on different datasets for the same phenotype, it is not clear what training means in this context. It appears that the relevant pathways are identified on the training dataset but the iPANDA scores for samples on the test set still use statistical weights derived by comparing cases and controls on the test set case in which the performance will be obviously optimistically biased. A particular limitation of pathway based approaches is that there may not be good pathways for a given phenotype, or even if there are, only a proportion of genes in the pathway may show useful discrimination signal. Therefore, no matter the approach to reduce the dimensionality of the gene set, the overall signal may still be noisy compared to using a few most predictive genes regardless in what pathway they are. The authors used $p < 0.05$ gene selection, but best approaches in the field use cross-validation to determine how many genes to use and may use a threshold on the magnitude of change to obtain robust models (PMID:23966112, PMID: 24994890). The performance of such methods should be presented next to the data in Table 1 so that one can judge the usefulness of the approach for class prediction.

3) The methods used to compare iPANDA against, GSEA and SPIA were designed to compare two groups of samples and not perform single sample pathway activation calculations. Specifically GSEA requires multiple samples to assess significance of normalized enrichment scores by permutations.

Single-sample GSEA is an alternative method that could have been used, but the original GSEA that appears to have been used herein cannot be used with one sample. For SPIA, the total accumulation tA statistic cannot be used as such to infer pathway activity in one sample since permutations of genes on the pathway are needed to evaluate the significance of pathway perturbation score (tA). Moreover, the second component used in SPIA comes from hypergeometric enrichment which again requires two groups of samples and assessment of differential expression using appropriate methods. Therefore, as it stands this paper does not convince this reviewer that iPANDA is better than existing pathway analysis methods or than existing outcome prediction methods using omics data.

4) There is also an incompatibility in the design of the iPANDA for pathway analysis and outcome prediction. This is because the statistical weights involved in computing iPANDA scores make use differential expression evidence (p-values from t-tests) derived from the full set of cases and control samples in the given experiment. While the resulting pathway scores may be used to rank pathways that are relevant for the current dataset, they cannot be used to judge the pathway score ability to discriminate between cases and controls on the current dataset because of the obvious over-fitting that will occur. This is like finding a set of genes that show most differential expression between cases and controls and then be surprised that an average of normalized expression levels (with respect to the mean in controls) for the top genes (since they will have highest statistical weights) does discriminate between cases and controls.

Reviewer #2 (Remarks to the Author):

A. Summary of the key results

The paper suggests a novel efficient method for transcriptomic data analysis on the signaling pathway level. This method, called iPANDA extends on the popular OncoFinder algorithm and can be used to identify pathways relevant to the specific biological conditions as well as for the discrimination between various biological or clinical conditions. According to the manuscript, iPANDA outperforms many popular pathway analysis methods. Breast cancer chemo pre-treatment data with the known outcome is chosen to demonstrate the capabilities of the method.

The paper is well written, the design of performed computational experiments is clear and described extensively in the text. The validation pipeline is comprehensive. The approach using coexpression statistics data from the CoexpressDB database for network topology treatment should be explicitly highlighted as a very good solution which was not implemented before. There is no doubt that suggested method can be extremely useful and at least worth trying when solving complex problems associated with transcriptomic data analysis. The algorithm seems to be superior to the popular, but commercial OncoFinder and the authors are making it freely available.

B. Originality and interest: if not novel, please give references

The paper is original and the approach to pathway activation analysis is novel. I was most curious

about the topological weight. The original concept of the topological weight for gene networks was proposed in 2005 by Hon Nian Chua, Wing-Kin Sung and Limsoon Wong "Exploiting indirect neighbours and topological weight to predict protein function from protein-protein interactions". However, they used a different formula. It would be interesting to see how/why the proposed formula differs.

C. Data & methodology: validity of approach, quality of data, quality of presentation

I have a minor comment about the quality of presentation. Plots appear to be made in different styles using different software and font. It would be nice if they could be made in the same style.

Also, the "Guide to authors" states that "The main text of an Article should begin with an introduction (without heading) of referenced text that expands on the background of the work (some overlap with the abstract is acceptable)". It seems that the authors need to remove the heading "Challenges of the modern transcriptomic data analysis".

I would recommend the paper to be proof-read by a native English speaker.

D. Appropriate use of statistics and treatment of uncertainties

The authors used sufficient sample sizes and made significant efforts to estimate the cross-study validity using separate training and test data sets

E. Conclusions: robustness, validity, reliability

The authors were careful and conservative in their conclusions.

F. Suggested improvements: experiments, data for possible revision

There are some points that should be essentially addressed before I can recommend this paper for publication:

1. The methodology used for comparison of iPANDA with other algorithms should be expanded to conclusively evaluate the performance of the algorithm. The authors claim that iPANDA can be used for identification of the signaling pathway networks relevant to a specific biological condition. SPIA and GSEA approaches are utilized for comparison with iPANDA method in the paper. Authors demonstrate that at least for the breast cancer data the pathways identified by iPANDA are more robust between various datasets and more clinically relevant according to the literature data than the pathway sets obtained using SPIA and GSEA. However, the comparison performed in the last subsection of the results shown in Figure 5 and Supplementary Figures 4 and 5 related to the application of pathway activation scores as classifiers between biological states does not seem fully appropriate. SPIA total accumulation values (tA) and GSEA enrichment scores (NES) are statistical measures of pathway perturbation, and they were not designed for the explicit discrimination between biological conditions. The reviewer recommends performing additional comparisons with

several methods specifically designed for such a purpose (e.g. MIPA or DART method, which demonstrated high efficiency in the classification of breast cancer types).

2. The only comparison with gene-level methods is performed on the MAQC-I dataset regarding noise reduction in the similar data acquired on various profiling platforms, and it looks highly convincing. However, the direct comparison with recent gene level methods (e.g. methods from the IMPROVER project) for biological/medical condition discrimination based on transcriptomic data would also be very useful. Such a comparison will make the overall perception of the novel algorithm much stronger and is highly recommended.

G. References: appropriate credit to previous work?

The authors state that "...remains difficult to achieve robust results over a group of independent data sets even when they are obtained from the same profiling platform..", which would require either a reference or an illustration.

The authors should consider citing

<https://bmcsystbiol.biomedcentral.com/articles/10.1186/s12918-016-0260-9>

H. Clarity and context: lucidity of abstract/summary, appropriateness of abstract, introduction and conclusions

The paper is well written and easy to read.

Reviewer #1 (Remarks to the Author):

The article presents a new pathway analysis method that allegedly is better at identifying pathways relevant to a given condition while the pathway summary scores can be used to phenotype prediction. The paper is well written and there is a substantial amount of work that was involved in this manuscript. However, there are several aspects that do not allow to evaluate the usefulness of the new approach as follows:

Traditionally pathway analysis methods were introduced to aid in interpreting differential gene expression results by identifying predefined functional categories or pathways that are perturbed in a given phenotype based on gene expression levels.

We thank the reviewer for the thoughtful review and comments.

1) **Although the authors claim that there is "no standardized pipeline for benchmarking transcriptomic data analysis algorithms", there are ways to assess whether a given pathway analysis method identifies biological pathways that are relevant to a given dataset as described elsewhere under minimal assumptions (PMID:25147207, PMID:24260172).**

Moreover, methods to develop prediction models from transcriptomics data have also been evaluated including large scale initiative such as MAQC-II and IMPROVER using independent datasets for evaluation of the predictive performance. The methods used for these two goals (pathway identification and prediction model development) are different, while the current paper presents iPANDA which is compared to existing methods that were designed for the first goal rather than the latter. So structurally the approach to benchmark the new method is not appropriate.

We agree with reviewers' notion that there are two distinct important issues which should be addressed when using pathway-based methods for gene expression data analysis: identification of relevant pathways for particular biological condition and prediction model development. Our method iPANDA was initially designed to address both of these issues. In the revised manuscript we demonstrate iPANDA performance in respect to both possible applications.

We thank the reviewer for suggesting two previously published studies describing benchmarking pathway analysis methods in respect to their ability to obtain pathways relevant for particular experimental case (specifically PMID:24260172 and PMID:25147207). We have rephrased the text and we are now citing these and other benchmarking approaches that are publicly available. To fully address the reviewer's concern, we have now assessed iPANDA performance in respect to prioritization criteria proposed in PMID:24260172, comparing iPANDA with the best performing methods described in papers suggested by the reviewer (PLAGE, PADOG, ORA and MIPA). The results of this comparison are shown in Supplementary figure 7. Even though iPANDA has lower performance than MIPA and PADOG in this test, it still gives meaningful results, particularly, the pathways expected to be perturbed have significantly lower ranks (higher scores). Moreover, iPANDA has similar performance to many common pathway analysis methods and outperforms some well known methods including ssGSEA and PLAGE (see Tarca et al., 2013).

Moreover, we have now performed additional comparisons of iPANDA using breast cancer data with several other methods, such as: ssGSEA (as recommended by this reviewer), DART (suggested by the second reviewer) and PLAGE (best performing method on benchmarking pipeline) in respect to their ability to find differential pathways with high AUC values. This experiment was held in a slightly different way from the benchmarking experiment proposed in PMID:24260172, as pathway significance scores for the whole group of case samples (responders and non-responders to paclitaxel treatment) were calculated against corresponding normal tissue samples. Sample-wise scores for each case sample were obtained using several pathway analysis

methods to assess the ability of these methods to obtain highly discriminative pathway scores between responders and non-responders groups. In contrast to the results from earlier prioritization benchmarking, iPANDA results in this latter test have outperformed all other methods (Figure 3, Supplementary figures 2-6).

Furthermore, we also performed additional validation to ensure the ability of iPANDA scores to be used as classifiers in the prediction models. Since iPANDA algorithm was specifically designed as an analytical tool for oncology data, the additional validation also included the assessment of iPANDA-based classifiers performance on the three cancer datasets (Breast Cancer, Multiple Myeloma and Neuroblastoma), derived from well known MicroArray Quality Control II (MAQC-II) study including ten independent endpoints (D,E,F,G,H,I,J,K,L,M). The performance results of our domestic random forest based classifier used with iPANDA scores as the input compared to external gene-level predictors developed during the MAQC-II project are described in the revised manuscript and shown in Supplementary Figure 8. Moreover, we are now showing the performance of the best classifier from IMPROVER project (as implemented in MaPredictDSC R package) for Breast cancer and Multiple myeloma endpoints. Unfortunately, it was not possible to use the same approach for Neuroblastoma dataset (which was acquired using Agilent profiling platform), as Agilent was not supported by the current MaPredictDSC method implementation. As it now shown on Supplementary Figure 8, iPANDA-based classifier outperforms other methods used for comparison on 3 out of 5 cancer-specific endpoints available for comparison. During our experiments we carefully followed the design of the original MAQC-II study protocol with separate training and validation datasets to obtain fair and comparable results.

Additionally, we have further benchmarked iPANDA with respect to another suggested hallmarks of pathway activation methods including noise reduction ability (using MAQC-I data) and ability to obtain robust results on the set of independent datasets related to the same biological condition (using breast cancer data). Taken together, we believe that thorough validation process of iPANDA algorithm described in the revised version of the manuscript is comprehensive and structurally convenient.

2) Although this reviewer is open to the idea of using pathway activation scores to improve outcome prediction performance, state of the art methods described in the MAQC-II and IMPROVER challenges, based on gene level predictors were not used as reference, and hence there is no reason why someone would believe that the new approach is worthwhile using.

To address reviewer's concern we have now performed a comprehensive comparison of iPANDA to state of the art methods described in the original MAQC-II experiment (overall best team and the best result for each of endpoint analyzed) regarding ten cancer endpoints (from D to M). Moreover, for endpoints D, E, F, G, H and I, we have compared our data to results generated with the best predictors from the IMPROVER DSC challenge. *Off note*, we were unable to use the IMPROVER code for neuroblastoma datasets used by the MAQC-II project, since these data was generated with Agilent profiling platform, whereas IMPROVER was designed to be used with Affymetrix.

Our random forest-based gene-level classifier demonstrates better prediction performance comparing to the average result of MAQC-II teams for the majority of endpoints. Notably, it outperforms MAQC-II candidate gene-level performance models for endpoints J and L. These results indicate the overall enhancement of machine-learning algorithms, since the MAQC experiment was performed 7 years ago, and suggest that our particular random-forest implementation can be utilized for further experiments.

The same random forest classifier trained on fold changes (FC) between samples under study and case samples, rather than on pure case samples gene expression, generates the same, or even better results for several "hard-to-predict" cancer endpoints (D,J and K). This observation demonstrates that incorporating expression data from corresponding normal tissue into the FC

calculation, can be a valuable addition when developing prediction models. In contrast, using the differentially expressed genes with fixed cutoff, failed to result in significant improvement in model prediction performance.

Furthermore, using the iPANDA scores for cancer-related pathways to train prediction models, allows to obtain significantly better results compared to the gene-level prediction models developed by either our team or MAQC-II consortium (for 3 out of 5 cancer endpoints available for comparison - F, J and K). The highest increase in performance was achieved on Neuroblastoma event free survival (endpoint K) (0.89 against 0.575 for the best MAQC-II team). These observations further support our notion that iPANDA algorithm can provide an efficient noise reduction when extracting biologically relevant features from the data.

Interestingly, while negative control endpoints (I and M) remained unpredictable for all our models, the MAQC-II positive control gender endpoints (H and L) have also appeared to be unpredictable when using models based on iPANDA scores for cancer-related pathways. These may be attributed to the fact that gender specific genes are poorly represented in cancer-related pathways utilized for prediction. Subsequently, our data indicates that performance of prediction models based on pathway-level data, greatly depends on the pathway database used.

The description of these additional experiments and corresponding figures have been now added to the revised version of the manuscript. Taken together, we believe that the addition of these data fully address the reviewer's concern.

* Although the authors present prediction performance results when training and testing on different datasets for the same phenotype, it is not clear what training means in this context. It appears that the relevant pathways are identified on the training dataset but the iPANDA scores for samples on the test set still use statistical weights derived by comparing cases and controls on the test set case in which the performance will be obviously optimistically biased.

As noted by the reviewer, we were indeed looking for relevant pathways for a certain biological phenotype (breast cancer) to distinguish between responders and non-responders. However, it has to be clarified that iPANDA scores were first calculated for all tumor vs normal samples (not for responders vs non-responders). And afterwards a set of discriminatory pathways between responders and non-responders group was identified. We apologize for the confusion. We have now rephrased the text to better explain the experiment. Moreover, in order to make training and validation procedure more straightforward and apparent, we have shifted from using the hierarchical clustering to random forest-based prediction models. Furthermore, since iPANDA calculates statistical weights using t-test for tumor compared to normal samples for each data set, we do not expect any bias caused by statistical weights as they all have been calculated for training and validation data sets independently.

* A particular limitation of pathway based approaches is that there may not be good pathways for a given phenotype, or even if there are, only a proportion of genes in the pathway may show useful discrimination signal.

We agree with the reviewer that discriminatory ability of pathway-based approaches may be hampered by the relevance of pathways database used for a certain biological condition, particularly in cases when only a proportion of genes in the pathway demonstrates useful discrimination signal. While the possibility that there may not be good pathways for a given phenotype cannot be completely eliminated, in this study we have used cancer pathways databases which contain a substantial portion of well-known and validated cancer-driving signaling axes common to most solid tumors. Therefore, we believe that at least in oncology

datasets, our algorithm will be able to define pathway activation signature useful for discrimination. Moreover, analysis of the comprehensive tumor pathway activation profile may be a more clinically relevant strategy to stratify the subset of patients whose tumors could probably respond and who would clinically benefit from anti-cancer therapeutic regimens than other outcome prediction methods based on the gene expression profile. While gene expression levels can be effectively used for phenotype prediction, it is quite possible that the most differentially expressed genes in a given signature won't be part of the pathways that actually drive tumor behavior. Alternatively, expression of some genes within the cancer driving pathways is not always predictive of the overall pathway activation. For example, low expression of epiregulin and amphiregulin is not a reliable indicator of EGFR pathway deactivation, which can be upregulated by activating mutations in downstream pathway targets, such as *BRAF*, *HRAS*, *NRAS*, *PI3K* and *AKT/PTEN*. Therefore, while there is currently no single flawless method for interpreting gene expression results, the proposed method of transcriptomic data analysis on the signaling pathway level may not only be useful for discrimination between various biological or clinical conditions, but may also aid in identifying functional categories or pathways that may be relevant as possible therapeutic targets.

Therefore, no matter the approach to reduce the dimensionality of the gene set, the overall signal may still be noisy compared to using a few most predictive genes regardless in what pathway they are.

The smooth thresholding approach described in the manuscript helps make the model robust. We assessed iPANDA pathway activation scores to differential gene set on MAQC-I data in respect to their ability to perform noise reduction while preserving biologically relevant features (Fig 2). The degree of similarity between same cell samples processed using various transcriptome profiling platforms were used for this analysis. According to the results described in the section "iPANDA is an effective tool for noise reduction in transcriptomic data" the similarity calculated using pathway activation values generated by the iPANDA algorithm significantly exceeds the one calculated using fold changes for the differentially expressed genes. This result implies that iPANDA algorithm effectively reduces the noise in input gene expression data.

* The authors used $p < 0.05$ gene selection, but best approaches in the field use cross-validation to determine how many genes to use and may use a threshold on the magnitude of change to obtain robust models (PMID:23966112, PMID: 24994890). The performance of such methods should be presented next to the data in Table 1 so that one can judge the usefulness of the approach for class prediction.

We agree with the reviewer that an arbitrarily chosen fixed threshold for p-values may cause undesirable instability in gene sets used for calculations and as a result reduce the overall performance of the model. For this reason we did not use arbitrary $p < 0.05$ gene selection in our pathway analysis approach but instead we used a smooth threshold function for p-values described in Methods section (Formula 5). High and low thresholds for p-values were obtained on a training data set via cross-validation approach. We have now mentioned this in the revised version of the manuscript. We have also used a threshold on the magnitude of change during the development of our algorithm, although the results regarding noise reduction ability of the algorithm and robustness of the pathway lists for similar datasets has decreased when such filters were applied.

3) The methods used to compare iPANDA against, GSEA and SPIA were designed to compare two groups of samples and not perform single sample pathway activation calculations.

As the reviewer noticed, SPIA and GSEA methods perform the comparison between two groups of samples to obtain pathway enrichment scores. The method used by iPANDA is identical in this regard. iPANDA utilizes t-test p-values calculated over the whole group of input samples compared to corresponding normal samples to estimate the statistical weights which are further used to obtain sample-wise pathway activation scores. However, the term 'sample-wise' may be confusive. In the current manuscript, a score that was obtained for a single case (e.g. tumor) sample versus a set of normal samples is termed as a 'sample-wise' score. Although we totally agree with the reviewer that SPIA and GSEA methods were initially designed for the relevant pathway assessment for a given condition rather than for phenotype prediction, we selected these methods for comparison as they are among the most cited and highly acclaimed algorithms in the community. Nevertheless, we expect that a good pathway analysis approach would produce robust scores, which should demonstrate a certain degree of discriminative power if the pathway database is chosen accordingly to the biological condition under study. Therefore, we presume that comparison to other methods, including SPIA and GSEA would be appropriate, even if they were not specifically designed for use in prediction models. However, only iPANDA scores used as input for Random forest classifier, demonstrate an increase in ability to discriminate between samples comparing to gene-level data. We believe that this result is due to the significant noise reduction achieved by simultaneous use of topological and statistical filtering of the input data in iPANDA algorithm.

* Specifically GSEA requires multiple samples to assess significance of normalized enrichment scores by permutations. Single-sample GSEA is an alternative method that could have been used, but the original GSEA that appears to have been used herein cannot be used with one sample.

According to the GSEA manual page (http://software.broadinstitute.org/gsea/doc/GSEAUUserGuideFrame.html?Run_GSEA_Page) and GSEA FAQ page (http://www.broadinstitute.org/cancer/software/gsea/wiki/index.php/FAQ#Can_I_use_GSEA_to_analyze_a_dataset_that_contains_a_single_sample.3F) GSEA can be run on the dataset containing a single case sample versus a set of normals. In this case, GSEA has no way of ranking the genes in such dataset. Nevertheless, genes can be ranked prior to the actual sample-wise score calculation using the whole set of samples. Hence, it is possible to obtain single sample pathway activation scores using the guidelines given in the GSEA manual. Another approach is to use permutation type set to 'gene_set' and lower FDR cutoff (5%) as suggested in the manual. To perform calculations the second method was applied. The Methods section in iPANDA manuscript describing GSEA settings used for comparison to iPANDA was rephrased and extended to make the whole pipeline of GSEA scores computation clear.

* For SPIA, the total accumulation tA statistic cannot be used as such to infer pathway activity in one sample since permutations of genes on the pathway are needed to evaluate the significance of pathway perturbation score (tA). Moreover, the second component used in SPIA comes from hypergeometric enrichment which again requires two groups of samples and assessment of differential expression using appropriate methods.

Differential expression for each case sample versus a set of normal samples can be used as an input for SPIA to obtain sample-wise scores for case samples. So the permutations of genes as well as assessment of differential expression are possible when using such design for the SPIA calculation. However, we agree that it is arguable whether it is fully appropriate to use SPIA total

accumulation as the input for phenotype prediction methods, which we have mentioned explicitly in the manuscript.

* Therefore, as it stands this paper does not convince this reviewer that iPANDA is better than existing pathway analysis methods or than existing outcome prediction methods using omics data.

To address reviewer's concern, we have now performed additional comparisons with ssGSEA, DART and PLAGE pathway analysis methods. As demonstrated on Figure 5, in contrast to iPANDA scores, pathway perturbation scores obtained using these methods are insufficient when using them as input for further phenotype prediction models. Moreover, comparisons with gene-level methods from MAQC-II and IMPROVER challenges were performed in respect to the ability to discriminate endpoints from MAQC-II datasets. Based on the results presented in the revised manuscript, we believe that iPANDA is superior to existing pathway analysis methods, and may be also used as a part of successful phenotype prediction pipeline.

4) There is also an incompatibility in the design of the iPANDA for pathway analysis and outcome prediction. This is because the statistical weights involved in computing iPANDA scores make use differential expression evidence (p-values from t-tests) derived from the full set of cases and control samples in the given experiment.

While the resulting pathway scores may be used to rank pathways that are relevant for the current dataset, they cannot be used to judge the pathway score ability to discriminate between cases and controls on the current dataset because of the obvious over-fitting that will occur.

This is like finding a set of genes that show most differential expression between cases and controls and then be surprised that an average of normalized expression levels (with respect to the mean in controls) for the top genes (since they will have highest statistical weights) does discriminate between cases and controls.

The reviewer has correctly noticed that we used statistical weights to compute iPANDA scores for each data set, and we explained earlier that statistical weights are calculated independently for each experiment based on t-test p-values. We would like to highlight the fact that iPANDA scores show the pathway activation profile of tumor vs normal comparison (not of responders vs non-responders). Subsequently, a set of top 30 discriminatory pathways between responders and non-responders was found in each training data set and then these sets of pathways were used for prediction on the test data sets. To strengthen and clarify the process of prediction model training and validation, in the revised version of the manuscript we have developed random forest-based prediction models instead of hierarchical clustering, so the whole data without prior feature selection was passed to the classifier in this case.

We agree with the reviewer that we found a set of genes most differentially expression between tumor and normal tissue samples, however, our experiments are not tuned in any way to select specific genes with high impact on treatment response. It is likely that among all differential genes derived from comparison of tumor and normal samples, the genes relevant to discrimination between responders and non-responders are also obtained. It is quite possible that the genes responsible for breast cancer phenotype are also playing essential role in treatment response.

Reviewer #2 (Remarks to the Author):

A. Summary of the key results

The paper suggests a novel efficient method for transcriptomic data analysis on the signaling pathway level. This method, called iPANDA extends on the popular OncoFinder algorithm and can be used to identify pathways relevant to the specific biological conditions as well as for the discrimination between various biological or clinical conditions. According to the manuscript, iPANDA outperforms many popular pathway analysis methods. Breast cancer chemo pre-treatment data with the known outcome is chosen to demonstrate the capabilities of the method.

The paper is well written, the design of performed computational experiments is clear and described extensively in the text. The validation pipeline is comprehensive. The approach using coexpression statistics data from the CoexpressDB database for network topology treatment should be explicitly highlighted as a very good solution which was not implemented before. There is no doubt that suggested method can be extremely useful and at least worth trying when solving complex problems associated with transcriptomic data analysis. The algorithm seems to be superior to the popular, but commercial OncoFinder and the authors are making it freely available.

B. Originality and interest: if not novel, please give references

* The paper is original and the approach to pathway activation analysis is novel. I was most curious about the topological weight. The original concept of the topological weight for gene networks was proposed in 2005 by Hon Nian Chua, Wing-Kin Sung and Limsoon Wong "Exploiting indirect neighbours and topological weight to predict protein function from protein-protein interactions". However, they used a different formula. It would be interesting to see how/why the proposed formula differs.

We thank the reviewer for his/her question. As correctly noted by the reviewer, we've utilized different formula to calculate topological weights for genes on a pathway network. This discrepancy derives from the different endpoints of the analyses. While Chua and coauthors aimed to identify genes with similar function using pathway topology information, the endpoint of our method is to assess the perturbation of a given signalling pathway as a whole. We are now citing the above studies in the revised version of the manuscript.

C. Data & methodology: validity of approach, quality of data, quality of presentation

* I have a minor comment about the quality of presentation. Plots appear to be made in different styles using different software and font. It would be nice if they could be made in the same style.

While we agree that the data presentation style slightly varies between the figures, we have created our figures with an attempt to make visual representation of the data as less bewildering to the reader as possible, while providing all necessary information related to the presented results. Nevertheless, please note that the figures style and their layout has been modified in the revised manuscript.

* Also, the "Guide to authors" states that "The main text of an Article should begin with an introduction (without heading) of referenced text that expands on the background of the work (some overlap with the abstract is acceptable)". It seems that the authors need to remove the heading "Challenges of the modern transcriptomic data analysis".

As recommended by the reviewer, we have removed the above heading from the revised version of the manuscript.

* I would recommend the paper to be proof-read by a native English speaker.

As recommended by the reviewer, the revised text has been proofed for language and style by the native English speaker.

D. Appropriate use of statistics and treatment of uncertainties

The authors used sufficient sample sizes and made significant efforts to estimate the cross-study validity using separate training and test data sets

E. Conclusions: robustness, validity, reliability

The authors were careful and conservative in their conclusions.

F. Suggested improvements: experiments, data for possible revision

G. References: appropriate credit to previous work?

* The authors state that "...remains difficult to achieve robust results over a group of independent data sets even when they are obtained from the same profiling platform..", which would require either a reference or an illustration. The authors should consider citing:

<https://bmcsystbiol.biomedcentral.com/articles/10.1186/s12918-016-0260-9>

The recommended manuscript is now cited in the revised version of our work.

H. Clarity and context: lucidity of abstract/summary, appropriateness of abstract, introduction and conclusions

The paper is well written and easy to read.

There are some points that should be essentially addressed before I can recommend this paper for publication:

We thank the reviewer for the detailed and thoughtful review.

1. The methodology used for comparison of iPANDA with other algorithms should be expanded to conclusively evaluate the performance of the algorithm. The authors claim that iPANDA can be used for identification of the signaling pathway networks relevant to a specific biological condition.

We thank the reviewer for his/her note. We have significantly reworked methodology in the revised manuscript, and it now includes a detailed description of all the comparisons to other pathway-based approaches (such as MIPA, DART and ssGSEA) and gene level methods (such as from MAQC-II experiment and IMPROVER project).

* SPIA and GSEA approaches are utilized for comparison with iPANDA method in the paper. Authors demonstrate that at least for the breast cancer data the pathways identified by iPANDA are more robust between various datasets and more clinically relevant according to the literature data than the pathway sets obtained using SPIA and GSEA. However, the comparison performed in the last subsection of the results shown in Figure 5 and Supplementary Figures 4 and 5 related to the application of pathway activation scores as classifiers between biological states does not seem fully appropriate. SPIA total accumulation values (tA) and GSEA enrichment scores (NES) are statistical measures of pathway perturbation, and they were not designed for the explicit discrimination between biological conditions.

As the reviewer correctly noticed, SPIA and GSEA methods perform the comparison between two groups of samples to obtain pathway enrichment scores. The method used by iPANDA is identical in this regard. Although we totally agree with the reviewer that SPIA and GSEA methods were initially designed for the relevant pathway assessment for a given condition rather than for phenotype prediction, we selected these methods for comparison as they are among the most cited and highly acclaimed algorithms in the community. Nevertheless, we expect that a good pathway analysis approach would produce robust scores, which should demonstrate a certain degree of discriminative power if the pathway database is chosen accordingly to the biological

condition under study. Therefore, we presume that comparison to other methods, including SPIA and GSEA would be appropriate, even if they were not specifically designed for use in prediction models. However, only iPANDA scores used as input for Random forest classifier, demonstrate an increase in ability to discriminate between samples comparing to gene-level data. We believe that this result is due to the significant noise reduction achieved by simultaneous use of topological and statistical filtering of the input data in iPANDA algorithm.

Differential expression for each case sample versus a set of normal samples can be used as an input for SPIA to obtain sample-wise scores for case samples. So the permutations of genes as well as assessment of differential expression are possible when using such design for the SPIA calculation. However, we agree with the reviewer that it is arguable whether it is fully appropriate to use SPIA total accumulation as the input for phenotype prediction methods. This is clearly stated in the revised version of the manuscript.

* The reviewer recommends performing additional comparisons with several methods specifically designed for such a purpose (e.g. MIPA or DART method, which demonstrated high efficiency in the classification of breast cancer types).

As recommended by the reviewer, we have compared the iPANDA performance with that of MIPA and DART methods, and the results of these experiments is now described in the revised manuscript. In contrast to iPANDA scores, pathway perturbation scores obtained by MIPA, DART and ssGSEA are also insufficient when using them as input for further phenotype prediction models. Moreover, comparisons with gene-level methods from MAQC-II and IMPROVER challenges were performed in respect to the ability to discriminate endpoints from MAQC-II datasets. Based on the results presented in the revised manuscript, we believe that on average, iPANDA is superior to the existing pathway analysis methods, and may also be used as a part of successful phenotype prediction pipeline.

2. The only comparison with gene-level methods is performed on the MAQC-I dataset regarding noise reduction in the similar data acquired on various profiling platforms, and it looks highly convincing. However, the direct comparison with recent gene level methods (e.g. methods from the IMPROVER project) for biological/medical condition discrimination based on transcriptomic data would also be very useful. Such a comparison will make the overall perception of the novel algorithm much stronger and is highly recommended.

To address reviewer's concern we have now performed a comprehensive comparison of iPANDA to the state-of-the-art methods described in the original MAQC-II experiment (overall best team and the best result for each of endpoint analyzed). Moreover, for endpoints D, E, F, G, H and I, we have compared our data with results generated with the best predictors from the IMPROVER DSC challenge. *Off note*, we were unable to use the IMPROVER code for neuroblastoma datasets used by the MAQC-II project, since these data was generated with Agilent profiling platform, whereas IMPROVER was designed to be used with Affymetrix.

Our random forest-based gene-level classifier demonstrates better prediction performance compared to the average result of MAQC-II teams for majority of the endpoints. Notably, it outperforms MAQC-II candidate gene-level performance models for endpoints J and L. These results indicate the overall enhancement of machine-learning algorithms, since the MAQC experiment was performed seven years ago, and suggest that our particular random-forest implementation can be utilized for further experiments.

The same random forest classifier trained on fold changes (FC) between samples under study and case samples, rather than on pure case samples gene expression, generates the same, or even better results for several “hard-to-predict” cancer endpoints (D,J and K). This observation demonstrates that incorporating expression data from corresponding normal tissue into the FC calculation, can be a valuable addition when developing prediction models. In contrast, using the differentially expressed genes with fixed cutoff, does not result in significant improvement in model prediction performance. Furthermore, using the iPANDA scores for cancer-related pathways to train prediction models, allows to obtain significantly better results compared to the gene-level prediction models developed by either our team or MAQC-II consortium (for 3 out of 5 cancer endpoints available for comparison F,J and K). The highest increase in performance was achieved on Neuroblastoma event free survival (endpoint K) (0.89 against 0.575 for the best MAQC-II team). These observations further support our notion that iPANDA algorithm can provide an efficient noise reduction when extracting biologically relevant features from the data.

Interestingly, while negative control endpoints (I and M) remained unpredictable for all our models, the MAQC-II positive control gender endpoints (H and L) have also appeared to be unpredictable when using models based on iPANDA scores for cancer-related pathways. These may be attributed to the fact that gender specific genes are poorly represented in cancer-related pathways utilized for prediction. Subsequently, our data indicates that performance of prediction models based on pathway-level data, greatly depends on the pathway database used.

A detailed description and discussion of these additional experiments have been now added to the revised version of the manuscript.

The table below is for reference purposes only and describes our efforts to compile and reproduce the various pipelines to perform the comparative analysis. While we received excellent support from several groups developing pathway analysis pipelines, many of the pipelines were not available.

Method	PMID	Code available	Similar data to compare	Online system for testing	Request for code sent	Last request	Response received	Code work on the shelf	Able to replicate the results	Referenced/ cited in the manuscript
ssGSEA	19847166	Yes	No	GSVA package	No			Yes		Yes
PLAGE	16156896	Yes	No	GSVA package	No			Yes		Yes
MIPA	25147207	No	Yes	No	7/12/2016	8/20/2016	Yes	No	Yes	Yes
DART	22011170	Yes	No	DART R package	No			Yes		Yes
maPredict DSC	23966112	Yes	Yes	maPredict DSC R package	No			Yes		Yes

Reviewer #1 (Remarks to the Author):

The authors attempted to respond to all my previous comments and have went to a great length to address my concerns by using additional datasets and methods in their comparative study. After giving this paper a fresh look I believe that the evidence presented is not convincing that iPANDA is a game changer in pathway analysis to be published in this journal as research of broad interest.

My major points are:

1) The first result shown in the paper is for the claim that iPANDA reduces noise in transcriptomics data. The authors compare the cross-platform mRNA abundance correlations of same biological samples for 1) iPANDA derived pathway scores 2) for individual genes with $p < 0.05$ and 3) for pathway scores obtained with other pathway analysis methods. Methods are considered better if the correlations achieved are higher. A limitation of this analysis is the fact that it is biased in favor of iPANDA since iPANDA gives more weight to genes that tend to be reliably co-expressed using information from COEXPRESSdb database. Reasons for genes to have higher correlation across multiple datasets studying different phenotypes on different platforms can be that these genes are in general more abundant (and hence more likely to be accurately measured on multiple platforms), and that they changed in expression across previously studied phenotypes. Therefore information from genes that change in new phenotypes and that are expressed at lower levels in general will be omitted, hence possibly limiting future discoveries.

2) The second piece of evidence provided for the usefulness of iPANDA is that the pathway scores are allegedly better biomarkers than either individual genes or than pathway summaries obtained with other methods. Figure 3 shows stable pathway scores for iPANDA when comparing responders to non-responder to paclitaxel treatment across multiple breast cancer datasets while this is not some much the case for the competitor methods (supplementary figures 2-6). Yet, the weights of genes in iPANDA are tuned using information these particular datasets: "T-test p-values for genes were calculated over the whole group of breast cancer samples against healthy tissue samples in order to estimate the statistical weights, which were further used to obtain sample-wise pathway activation iPANDA scores. Cross-validation approach using samples from GSE20194 data set was utilized to obtain the threshold values for calculation of statistical weights and merging the gene modules." Although in their response the authors mention "However, it has to be clarified that iPANDA scores were first calculated for all tumor vs normal samples (not for responders vs non-responders)" the reader is left to trust that there is no possible information leak from one phenotype to the other. In contrast all the other methods are not tuned in any way to these particular datasets. This fact makes also the appreciation of classification results in Figure 5 difficult.

3) The third type of evidence for the applicability of iPANDA is comparing the ranks of putatively relevant pathways over multiple datasets, benchmark proposed previously and used by others as well. Based on this assessment, iPANDA is no better than a hypergeometric test (aka fisher's exact test) (method ORA in Supplementary Figure 7). So based on this assessment iPANDA is not an important advance in the field that the general readership of this journal should be aware of.

4) The use of the total net accumulation perturbation (tA) values for each pathway as a proxy for pathway activation for SPIA method is not appropriate as previously mentioned by this and the other reviewer. There are several reasons for this: 1) tA is only half of the evidence that SPIA considers and is independent on the information about the overall amount of differential expression

in the pathway that the other methods use. By design tA is independent on the number of differentially expressed genes but it cares of where in the pathways those genes are and what signs they have, and other factors. 2) tA is based on information derived only from the genes in the pathway that are connected with certain type of relations documented in KEGG. All other methods use information from all genes in the pathway, so the comparison is not fair.

Reviewer #2 (Remarks to the Author):

The authors have significantly improved the manuscript based on the reviewers' suggestions. The manuscript is now much easier to read, flow of information is smooth and ideas are presented in a clear and concise manner.

The authors performed the requested additional tests and clearly demonstrated the advantages and limitations of their approach. The results will be of interest to the broad scientific audience.

Manuscript NCOMMS-16-05573A. "In silico Pathway Activation Network Decomposition Analysis (iPANDA) as a method for biomarker development"

Reviewer #1 (Remarks to the Author):

The authors attempted to respond to all my previous comments and have went to a great length to address my concerns by using additional datasets and methods in their comparative study. After giving this paper a fresh look I believe that the evidence presented is not convincing that iPANDA is a game changer in pathway analysis to be published in this journal as research of broad interest.

My major points are:

1) The first result shown in the paper is for the claim that iPANDA reduces noise in transcriptomics data. The authors compare the cross-platform mRNA abundance correlations of same biological samples for

- 1) iPANDA derived pathway scores
- 2) for individual genes with $p < 0.05$ and
- 3) for pathway scores obtained with other pathway analysis methods.

Methods are considered better if the correlations achieved are higher. A limitation of this analysis is the fact that it is biased in favor of iPANDA since iPANDA gives more weight to genes that tend to be reliably co-expressed using information from COEXPRESSdb database. Reasons for genes to have higher correlation across multiple datasets studying different phenotypes on different platforms can be that these genes are in general more abundant (and hence more likely to be accurately measured on multiple platforms), and that they changed in expression across previously studied phenotypes. Therefore information from genes that change in new phenotypes and that are expressed at lower levels in general will be omitted, hence possibly limiting future discoveries.

As correctly noted by the reviewer, the iPANDA algorithm utilizes the information about correlation between genes obtained from the COEXPRESSdb database. However, iPANDA algorithm does not assign more weight to genes that tend to be reliably co-expressed. The information from COEXPRESSdb is utilized solely for grouping genes into modules. This key feature of the proposed approach could be considered advantageous over the platforms that solely rely on gene enrichment statistics, since according to our analysis this feature allows to generate a more robust signaling pathway signature for the same phenotype across various datasets (please refer to Figure 4 of the original manuscript).

While it is possible that the co-expression data for some low abundant genes may be indeed missing from the current version of the COEXPRESS db, genes whose expression levels are on the edge of detection by modern high throughput gene expression technologies will most likely be discarded from consideration by any transcriptomic analysis and most probably will not have any significant impact on cross-platform mRNA abundance correlations.

In support of these claims, below we are attaching the figure (similar to those presented in Figure 2 and Supplementary Figure 1 of the manuscript) where feature for grouping genes into

modules is now “switched off”, meaning that all genes are considered individually and no information from COEXPRESSdb is being utilized. This result clearly demonstrates that iPANDA scores obtained even with the gene modules feature “switched off” show higher sample-wise similarity between data obtained using various profiling platforms compared to the similarity calculated on the gene level.

The Reviewer’s viewpoint on the usage of COEXPRESSdb for pathway analysis is somewhat unclear. On one hand, the Reviewer mentions that integration of the COEXPRESSdb data potentially improves the results of signalling pathway analysis. Whereas on the other hand, the Reviewer considers the usage of this publicly available data disadvantageous. Nevertheless, it is reasonable to assume that inclusion of publicly available data which improves the results and is not directly related to the datasets under consideration should be generally viewed as advantageous.

2) The second piece of evidence provided for the usefulness of iPANDA is that the pathway scores are allegedly better biomarkers than either individual genes or than pathway summaries obtained with other methods. Figure 3 shows stable pathway scores for iPANDA when comparing responders to non-responder to paclitaxel treatment across multiple breast cancer datasets while this is not some much the case for the competitor methods (Supplementary Figures 2-6). Yet, the weights of genes in iPANDA are tuned using information these particular datasets: “T-test p-values for genes were calculated over the whole group of breast cancer samples against healthy tissue samples in order to estimate the statistical weights, which were further used to obtain sample-wise pathway activation iPANDA scores. Cross-validation approach using samples from GSE20194 data set was utilized to obtain the threshold values for calculation of statistical weights and merging the gene modules.”

Although the Reviewer correctly noted that the threshold values for statistical weights were obtained using GSE20194 dataset (which was used only in training sets for phenotype prediction task), the gene weights in iPANDA are calculated for each of the datasets used for analysis separately (independently), and therefore, introduction of the artificial inter-datasets bias for the gene weights as a result of this approach is virtually impossible.

Although in their response the authors mention “However, it has to be clarified that iPANDA scores were first calculated for all tumor vs normal samples (not for responders vs non-responders)” the reader is left to trust that there is no possible information leak from one phenotype to the other. In contrast all the other methods are not tuned in any way to these particular datasets. This fact makes also the appreciation of classification results in Figure 5 difficult.

It is important to mention that no phenotype tuning was involved in data generation. The patients’ clinical outcome used for benchmarking (response to paclitaxel) was not exposed to iPANDA in any way, and therefore it could not be leaked. The initial method for iPANDA-based prediction models performance assessment was designed in a way which precludes any leakage across different phenotypes. To assure that the results presented in Figure 5 are not biased by possible information leak from one phenotype to another, the normalized iPANDA scores were calculated for all samples in each one of the six data sets separately for paclitaxel treatment response prediction procedure. Moreover, it is crucial to highlight that the same p-value thresholds were used for MAQC-II endpoint prediction. The results obtained using iPANDA-based models for MAQC-II endpoints, poses iPANDA as an efficient tool for development of prediction models whose application is not restricted only to breast cancer data.

3) The third type of evidence for the applicability of iPANDA is comparing the ranks of putatively relevant pathways over multiple datasets, benchmark proposed previously and used by others as well. Based on this assessment, iPANDA is no better than a hypergeometric test (aka fisher’s exact test) (method ORA in Supplementary Figure 7). So based on this assessment iPANDA is not an important advance in the field that the general readership of this journal should be aware of.

To address the iPANDA superiority in respect to ranking of the putatively relevant pathways, we have precisely followed the benchmarking approach requested by this reviewer, and the results of these comparison are now summarised in the revised manuscript “...*although the iPANDA algorithm did not surpass the alternative methods which were reported to be the best according to the prioritization criteria (PADOG and MIPA) (Supplementary Fig. 7), it outperforms some other popular methods including ssGSEA and PLAGE, and demonstrates the ability to generate highly relevant results, since the pathways expected to be perturbed have significantly lower ranks (higher scores) than if it was expected by chance*”.

As reported in the original manuscript that describes the benchmarking pipeline [PMID: 24260172], the PADOG is the only pathway analysis algorithm that surpasses the ORA method in respect to its ability to assign higher ranks to pathways relevant to a given condition (Fig. 2 in PMID:24260172). MIPA, another algorithm which outperforms ORA (described in PMID: 25147207) is not publicly available to the community. Nevertheless, we have contacted the authors of this algorithm and requested to perform the comparison outlined in our manuscript (their contributors are now acknowledged in the paper). Unfortunately, the comparison proposed by the Reviewer relies on a very specific set of pathways expected to be perturbed under certain disease conditions. Each of these pathways consists of genes associated with multiple mechanisms of biological regulation, therefore these pathways contain several sparsely interconnected components. In contrast, iPANDA is specifically designed to address regulatory circuits with well-defined topology (e.g., mTOR pathway, AKT pathway, etc.).

Additionally, the metrics used in the proposed benchmarking pipeline are prone to some limitations. The robustness of the results is lower for the most precise pathway analysis methods, as the wrong prediction for a single dataset in the pipeline may significantly affect the final score of the method and leave the entire method out of the game.

Based on the thorough assessment of iPANDA outlined in our manuscript, the proposed algorithm provides an important advance in transcriptomics data analysis and has a potential to have a high impact in the field.

4) The use of the total net accumulation perturbation (tA) values for each pathway as a proxy for pathway activation for SPIA method is not appropriate as previously mentioned by this and the other reviewer. There are several reasons for this: 1) tA is only half of the evidence that SPIA considers and is independent on the information about the overall amount of differential expression in the pathway that the other methods use. By design tA is independent on the number of differentially expressed genes but it cares of where in the pathways those genes are and what signs they have, and other factors. 2) tA is based on information derived only from the genes in the pathway that are connected with certain type of relations documented in KEGG. All other methods use information from all genes in the pathway, so the comparison is not fair.

The only metrics which can be used for the comparison in case of SPIA are total accumulation (tA) values and combined probability (pG). Although we tried to utilize both metrics for our analysis, despite the limitations brought by the Reviewer, tA showed a better performance in similarity and phenotype prediction tests. While the usage of this metric for comparison is arguable (*as explicitly mentioned in the revised manuscript*), the SPIA tA values are the only metric besides the iPANDA scores, which surpasses gene-level cross-platform correlations between MAQC-I samples and demonstrates better performance when used in phenotype prediction models comparing to gene-level data.

Reviewer #2 (Remarks to the Author):

The authors have significantly improved the manuscript based on the reviewers' suggestions. The manuscript is now much easier to read, flow of information is smooth and ideas are presented in a clear and concise manner.

The authors performed the requested additional tests and clearly demonstrated the advantages and limitations of their approach. The results will be of interest to the broad scientific audience.

We thank the reviewer for the thoughtful advice and comments.